# Intestinal non-canonical NFκB signaling shapes the local and systemic immune response

Sadeesh K. Ramakrishnan[1], Huabing Zhang[2], Xiaoya Ma[2], Inkyung Jung[2], Andrew J. Schwartz[2], Daniel Triner[2], Samantha N. Devenport[2], Nupur K. Das[2], Xiang Xue [2], Melody Y. Zeng[3,4], Yinling Hu[5], Richard M. Mortensen[2], Joel K Greenson[3], Marilia Cascalho[6,7,8], Christiane E. Wobus[8], Justin A. Colacino [9,10], Gabriel Nunez[3,11], Liangyou Rui [2,4] & Yatrik M. Shah[2,4]

Microfold cells (M-cells) are specialized cells of the intestine that sample luminal microbiota and dietary antigens to educate the immune cells of the intestinal lymphoid follicles. The function of M-cells in systemic inflammatory responses are still unclear. Here we show that epithelial non-canonical NFκB signaling mediated by NFκB-inducing kinase (NIK) is highly active in intestinal lymphoid follicles, and is required for M-cell maintenance. Intestinal NIK signaling modulates M-cell differentiation and elicits both local and systemic IL-17A and IgA production. Importantly, intestinal NIK signaling is active in mouse models of colitis and patients with inflammatory bowel diseases; meanwhile, constitutive NIK signaling increases the susceptibility to inflammatory injury by inducing ectopic M-cell differentiation and a chronic increase of IL-17A. Our work thus defines an important function of non-canonical NFκB and M-cells in immune homeostasis, inflammation and polymicrobial sepsis.

[1] Department of Medicine, University of Pittsburgh, Pittsburgh, PA 15261, USA. [2] Department of Molecular & Integrative Physiology, University of Michigan, Michigan, MI 48109, USA. [3] Department of Pathology, University of Michigan, Ann Arbor, MI 48109, USA. [4] Internal Medicine, University of Michigan, Ann Arbor, MI 48109, USA. [5] Cancer and Inflammation Program, Center for Cancer Research, National Cancer Institute, National Institutes of Health, Frederick, MD 21702, USA. [6] Transplantation Biology, University of Michigan, Ann Arbor, MI 48109, USA. [7] Department of Surgery, University of Michigan, Ann Arbor, MI 48109, USA. [8] Department of Microbiology and Immunology, University of Michigan, Ann Arbor, MI 48109, USA. [9] Department of Environmental Health Sciences, University of Michigan, Ann Arbor, MI 48109, USA. [10] Department of Nutritional Sciences, University of Michigan, Ann Arbor, MI 48109, USA. [11] Comprehensive Cancer Center, University of Michigan, Ann Arbor, MI 48109, USA. Correspondence and requests for materials should be addressed to S.K.R. (email: ramaks@pitt.edu) or to Y.M.S. (email: shahy@umich.edu)

The intestinal epithelial cells maintain a protective barrier and are central in sensing and initiating a proper mucosal immune response following infection or injury[1]. Dysregulated host immune response against commensal microbiota initiates inflammatory diseases of the intestine[2]. Specialized intestinal epithelial cells called Microfold cells (M-cells) are localized to the luminal surface of the Peyer's patches and colon lymphoid follicles. M-cell provides direct contact of immune cells in the intestinal lymphoid follicle to dietary antigens and microbiota via trans-epithelial transport and thus play a critical role in the mucosal immune response. However, the mechanisms that are involved in M-cell maintenance and its role in local and systemic immune responses are not clear.

NFκB signaling is a key mediator of cytokine and chemokine transcription and can be divided into two broad pathways. In the classical pathway, tumor necrosis factor (TNF)α-activated Iκβ kinase (IKK) phosphorylates the inhibitory Iκβ (IKKβ) resulting in the nuclear translocation of NFκB and expression of NFκB target genes. The non-canonical pathway involves activation of NFκB inducing kinase (NIK), which leads to proteolytic processing of NFκB2 to p52[2]. Non-canonical NFκB pathway plays an essential role in diverse biological processes, including lymphoid organogenesis, osteoclast differentiation, and cell-autonomous functions in immune cells[3].

In intestinal epithelial cells, the classical NFκB pathway acts as a rheostatic transcription factor. Disruption or constitutive activation leads to inflammation and injury[4–6]. Recent studies demonstrate that mutations in *Map3k14* (the gene which encodes NIK) or the upstream negative regulators of the non-canonical NFκB pathway leads to autoimmune or inflammatory disorders[7,8]. Allen et al. demonstrated that nucleotide-binding domain and leucine-rich-repeat containing protein (NLRP)12-mediated inhibition of NIK protects against intestinal inflammation via a non-hematopoietic cell lineage[9,10]. However, an independent study using *Nlrp12*[−/−] mice show an increase in intestinal inflammation via hematopoietic cell lineage that is dependent on classical NFκB and ERK pathway[11]. Lastly, a whole body p52 knockout mice showed protection following intestinal injury[12]. Studies using global *Nik*[−/−] mice have shown that loss of NIK signaling decreases both intestinal and systemic humoral response due to a defect in B-cell germinal center development. *Nik*[−/−] mice do not develop intestinal Peyer's patches and exhibit loss of M-cells[13,14]. Thus, the role of NIK signaling in the intestinal epithelium compared to hematopoietic cells in the regulation of humoral response is not clear. NIK signaling is a highly sought after target in many cancers[15], and therefore defining the physiological relevance of non-canonical NFκB pathways is crucial.

The role of non-canonical NFκB signaling in intestinal homeostasis still remain unclear. Here we show that intestinal epithelial NIK is critical for M-cell maintenance and to elicit a protective IgA response. We also find that the intestinal epithelial NIK is essential for the local and systemic regulation of IL-17 in animal models of colitis and polymicrobial sepsis. To our surprise, constitutive activation of epithelial NIK also leads to enhanced inflammation and injury. Thus, our work delineates an essential rheostatic role of non-canonical NFκB signaling in intestinal homeostasis.

## Results

**Epithelial NIK is essential for the maintenance of M cells**. To determine the role of non-canonical NFκB signaling in epithelial and hematopoietic cells in colitis, bone marrow chimeras of wild type (WT) and global NIK knock out (*Nik*[−/−]) mice were treated with the colitis-inducing agent dextran sulfate sodium (DSS) for 7 days. *Nik*[−/−] chimeras generated with WT bone marrow had a significant decrease in body weight and colon length, compared to WT mice that received either WT or *Nik*[−/−] bone marrow (Fig. 1a, b). H&E analysis revealed a significant increase in the infiltration of inflammatory cells in the colon of *Nik*[−/−] chimeras generated with WT bone marrow compared to WT mice that received either WT or *Nik*[−/−] bone marrow (Fig. 1c). This suggests that NIK signaling in non-hematopoietic cells play a critical role in colitis.

To determine if epithelial NIK plays a role in colitis, mice with an intestinal epithelial-specific disruption of NIK were generated using Cre recombinase driven under the villin promoter (*Nik*[ΔIE]). Under steady state, no morphological difference was observed in the small intestine and colon of *Nik*[ΔIE] mice (Fig. 1d and Supplementary Fig. 1a-c). In the intestine, NIK signaling is elevated in the Peyer's patches (PP) and colon lymphoid follicles (Colon LF) compared to adjacent epithelial cells (Supplementary Fig. 1d, e). A recent study using enteroids from global *Nik*[−/−] mice demonstrated a significant decrease in M-cell differentiation[16]. M-cells are specialized antigen sampling cells present on the luminal surface of PP and colon LF. Consistent with previous reports[16,17], M-cells were decreased in the PP and Colon LF of *Nik*[ΔIE] mice as assessed by glycoprotein 2 (GP2) and M-cell specific alpha-1, 2-fucose using UEA1-lectin, and the mRNA levels of *Spib*, *Gp2* and *Anxa5* (Fig. 1e, f and Supplementary Fig. 1f-h). When antigen sampling was assessed using microbeads, we observed a significant decrease in the localization of microbeads in the Peyer's patches and colon LF of *Nik*[ΔIE] mice demonstrating a critical role of epithelial NIK signaling in antigen sampling (Fig. 1g). The decrease in humoral response in *Nik*[−/−] mice was attributed to the loss of NIK signaling in B-cells, rather than defective antigen sampling[13,14,16]. Here, we show that the loss of intestinal epithelial NIK signaling (*Nik*[ΔIE]) is sufficient to decrease gut humoral response (Fig. 1h). Non-canonical NFκB signaling is mediated by NIK activation of IKKα[18]. Loss of epithelial IKKα (*Ikkα*[ΔIE]) recapitulated the phenotype of *Nik*[ΔIE] mice such as a decrease in M-cells and fecal IgA levels (Supplementary Fig. 1i-k).

The turnover of epithelial lining occurs every 3–5 days. To address whether epithelial NIK signaling is essential for M-cell maintenance, we generated mice with temporal loss of NIK using tamoxifen-inducible Cre recombinase driven under the villin promoter (*Nik*[F/F;VilERT2Cre]; Fig. 1i). Interestingly, disruption of epithelial NIK for 2 weeks decreased the PP size and number (Supplementary Fig. 1l, m). qPCR analysis showed a significant decrease in M-cell markers both in the PP and colon LF (Fig. 1j, k). Memory B-cells are shown to be an important player in eliciting a humoral response; however, it is not clear whether continuous antigenic stimulation is required to maintain, activate and/or proliferate memory B-cells[19]. Temporal loss of epithelial NIK for 2 weeks is sufficient to decrease fecal IgA levels in *Nik*[F/F;VilERT2Cre] mice (Fig. 1l), suggesting that maintenance of epithelial NIK signaling is essential for the gut humoral response. Together, the data suggest that continuous epithelial NIK signaling is essential for maintenance of M-cells and eliciting gut IgA response.

**Epithelial NIK signaling protects against colitis**. Intestinal injury induced by DSS robustly increased the expression of NIK and p52 in colon epithelium (Fig. 2a). To assess the role of epithelial NIK in colitis, *Nik*[ΔIE] mice were treated with DSS. *Nik*[ΔIE] mice on DSS exhibited bloody diarrhea with a significant decrease in body weight and colon length (Fig. 2b, c). H&E staining revealed a significant loss of epithelial architecture with infiltration of inflammatory cells suggesting exacerbated colitis in *Nik*[ΔIE] mice (Fig. 2d). Moreover, the induction of p52 by DSS is lost in

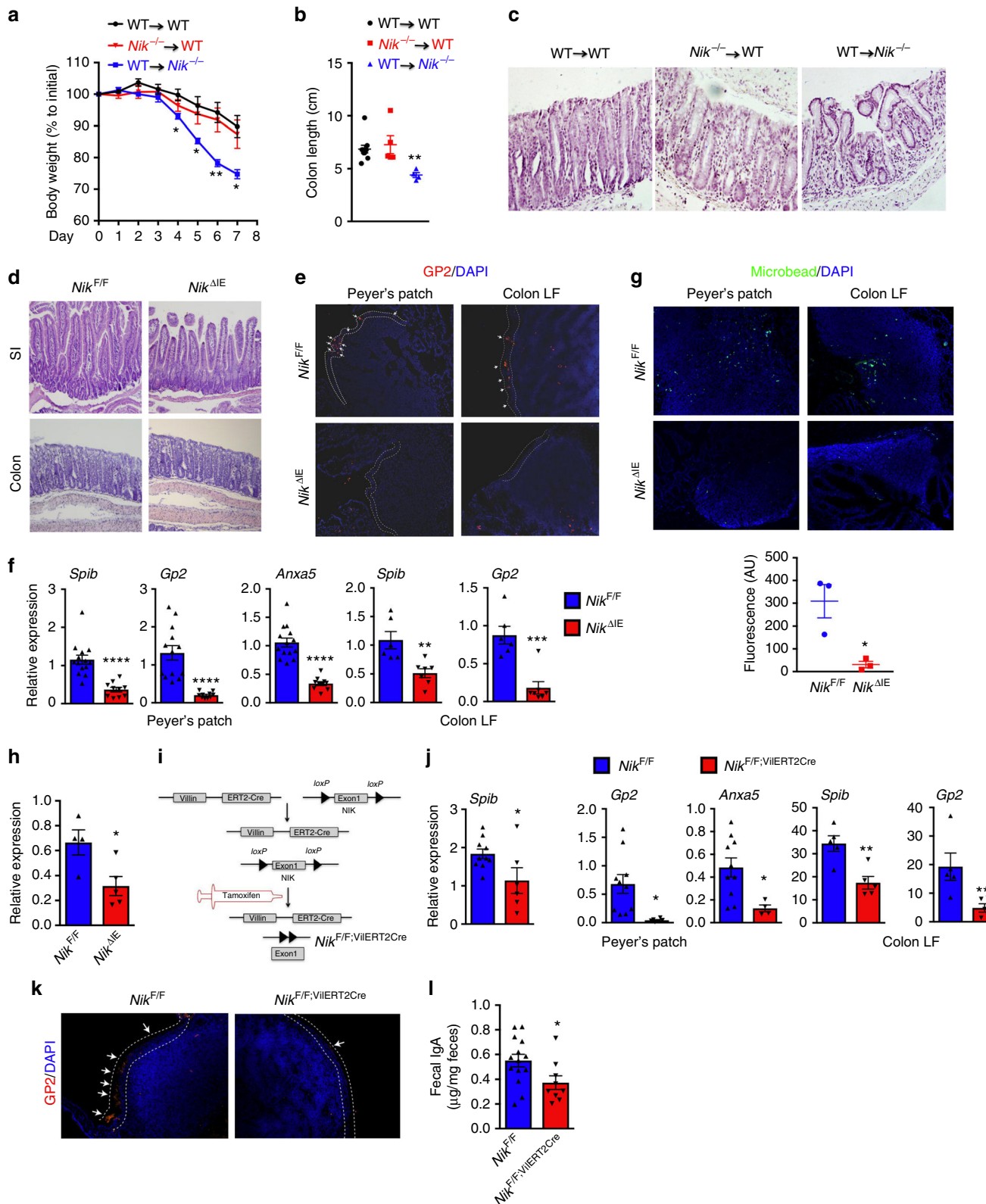

the colon of $Nik^{\Delta IE}$ mice (Fig. 2e). $Ikk\alpha^{\Delta IE}$ mice recapitulated the susceptibility to colitis similar to $Nik^{\Delta IE}$ mice (Supplementary Fig. 2a-d), suggesting that loss of epithelial non-canonical NFκB pathway increases colitis susceptibility.

We then questioned whether epithelial NIK regulates barrier function. Western blot analysis revealed no difference in the expression of key barrier function proteins such as occludin and E-cadherin in the colon of $Nik^{\Delta IE}$ mice (Supplementary Fig. 2e). Moreover, no difference in the barrier function was observed in DSS-treated $Nik^{\Delta IE}$ mice, assessed by serum FITC-dextran levels following oral gavage (Supplementary Fig. 2f). To determine if pro-inflammatory mediators are involved in the colitis susceptibility, we assessed the progression of DSS-induced colitis in a time-dependent manner. We noticed the histological difference

**Fig. 1** Epithelial NIK signaling is essential for M-cell maintenance. **a–c** Body weight (**a**), colon length (**b**), and H&E analysis (**c**) of WT and global NIK knock out ($Nik^{−/−}$) mice following bone marrow transplantation from WT or $Nik^{−/−}$ donors and treated with 2% DSS for 7 days; $n = 4$ mice/group. **d** H&E staining of small intestine (SI) and colon of mice with intestinal epithelial-specific disruption of NIK ($Nik^{\Delta IE}$) assessed at 1 month of age. Images were taken at ×10 magnification. **e** Staining for glycoprotein 2 (GP2) in the PP and colon LF of $Nik^{\Delta IE}$ mice. Images were taken at ×20 magnification. **f** qPCR analysis for M-cell markers in the Peyer's patches (upper panel) and colon LF (lower panel) of $Nik^{\Delta IE}$ mice. **g** Antigen sampling assessed in Peyer's patches using microbeads gavaged in $Nik^{\Delta IE}$ mice. Images were taken at ×20 magnification and fluoresence intensity was measured using ImageJ software. $n = 3$/group. **h** Fecal IgA assessed in 6–7-week-old $Nik^{\Delta IE}$ mice. **i** Schematic representation of mice with temporal disruption of intestinal epithelial NIK ($Nik^{F/F;VilERT2Cre}$) using tamoxifen inducible Cre recombinase driven by Villin promoter. **j** qPCR analysis for M-cell markers in the Peyer's patch and colon lymphoid follicles in $Nik^{F/F;VilERT2Cre}$ mice at 2 weeks after tamoxifen treatment. **k** GP2 staining in the Peyer's patches of $Nik^{F/F;VilERT2Cre}$ mice at 2 weeks after tamoxifen treatment. Images were taken at ×20 magnification. **l** Fecal IgA assessed in $Nik^{F/F;VilERT2Cre}$ mice at 2 weeks after tamoxifen treatment. Results are expressed as mean ± SEM. Significance determined using $t$ test. *$P < 0.05$; **$P < 0.01$; ***$P < 0.001$; ****$P < 0.0001$

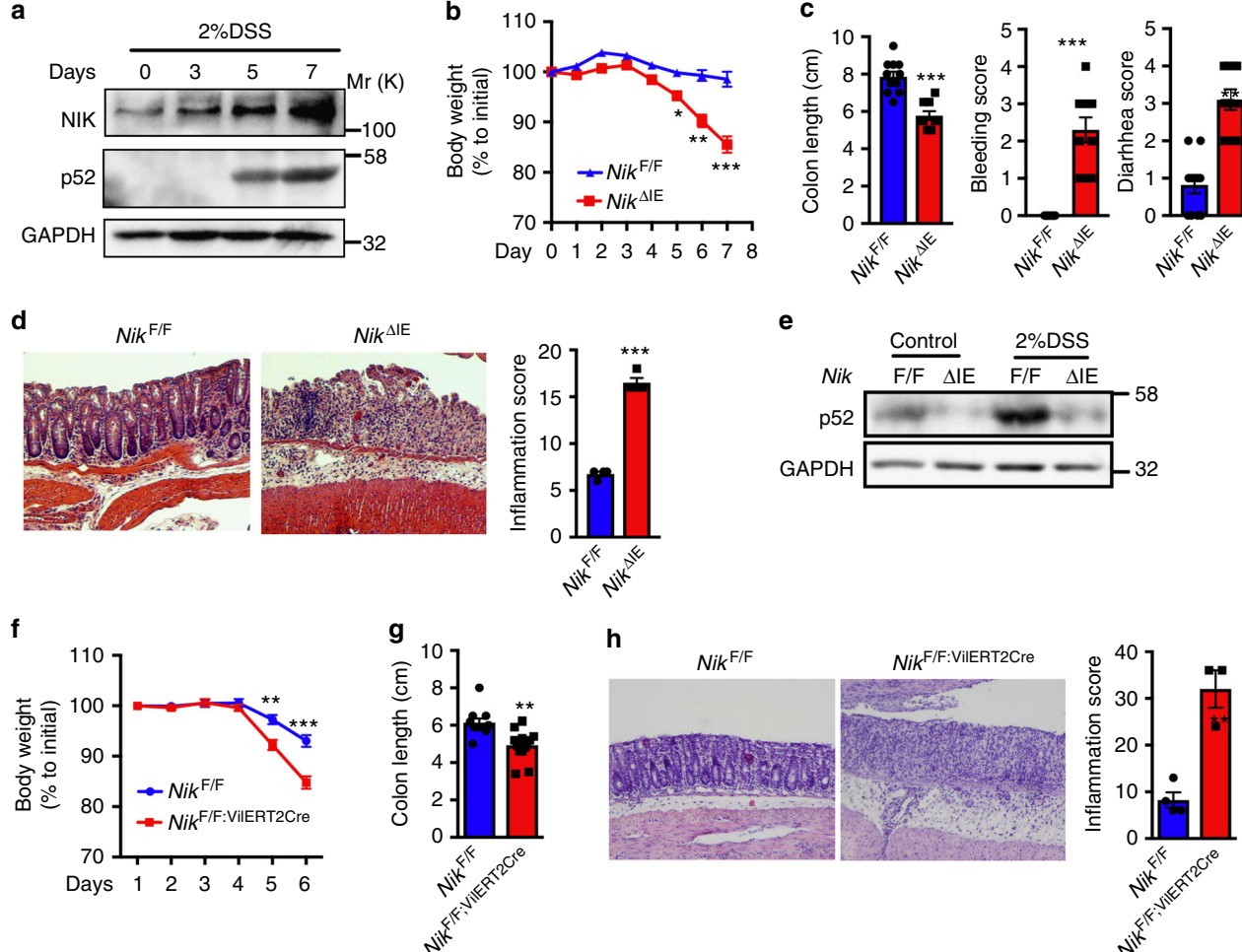

**Fig. 2** Epithelial NIK signaling protects against colitis. **a** Western blot analysis of NIK and p52 in the colon of WT mice treated with 2% DSS for 0–7 days. **b–d** Body weight (**b**), colon length, bleeding score, diarrhea score (**c**), and H&E analysis (**d**) in $Nik^{F/F}$ and $Nik^{\Delta IE}$ mice treated with 2% DSS for 7 days. **e** Western blot analysis of p52 in 6-week-old $Nik^{\Delta IE}$ treated with 2% DSS for 7 days. **f–h** Body weight (**f**), colon length (**g**), and H&E analysis (**h**) assessed in $Nik^{F/F}$ and $Nik^{F/F;VilERT2Cre}$ mice treated with 2% DSS for 6 days. H&E images were taken at × 10 magnification. Results are expressed as mean ± SEM. Significance determined using $t$ test. *$P < 0.05$; **$P < 0.01$; ***$P < 0.001$

only from day 5 following DSS treatment in $Nik^{\Delta IE}$ (Supplementary Fig. 2g). When we assessed for the pro-inflammatory mediators preceding histological changes i.e., 3 days of DSS treatment, no difference in the mRNA levels of $Tnf\alpha$, $Il6$, $Il1\beta$, and $Il10$ were noted in the colon of $Nik^{\Delta IE}$. However, at day 7 exacerbated expression of $Il6$ and $Il1\beta$ was observed in the colon correlating to the increase in histological injury in the $Nik^{\Delta IE}$ mice (Supplementary Fig. 2g-i), suggesting that pro-inflammatory mediators are not the early causative mechanism for colitis in

$Nik^{\Delta IE}$ mice. We further investigated if epithelial NIK has any role in a model of infectious colitis and radiation-induced injury. Loss of NIK exacerbates infectious colitis following *Salmonella typhimurium* SL1344 infection (Supplementary Fig. 2j, k); however, no difference in radiation-induced injury was observed (Supplementary Fig. 2l). $Nik^{F/F;VilERT2Cre}$ mice were also assessed to determine if a temporal loss of NIK will exacerbate colitis. DSS treatment significantly increased colitis in $Nik^{F/F;VilERT2Cre}$ mice as determined by body weight, colon length and inflammatory

cell infiltration in the colon by H&E (Fig. 2f–h). Together, the data demonstrate that basal intestinal epithelial NIK signaling is critical to protecting against colitis.

**Loss of epithelial RANK increases colitis susceptibility.** NIK signaling is induced by numerous ligands such as lymphotoxin (LT), RANKL, TWEAK, and CD40L. PP and LF are absent in mice with disruption of LTα, LTβ and LTβr[20,21], but no such loss of PP or LF has been reported in mice with disruption of TWEAK[22], suggesting that cellular cues differentially regulate M-cell differentiation and PP development. In primary duodenal enteroids from the wild-type mice, RANKL, but not Tweak or LTβ robustly induced p52 and M-cell markers (Fig. 3a–c and Supplementary Fig. 3a). Further, loss of epithelial NIK abrogated RANKL-mediated M-cell differentiation in duodenal and colon enteroids[17,23], but no difference in inflammatory or epithelial or secretory markers was noted (Fig. 3d and Supplementary Fig. 3b–e). To investigate if the loss of M-cells increases colitis susceptibility, mice with intestinal epithelial-specific disruption of the RANKL receptor were generated ($Rank^{ΔIE}$). Consistent with the role of RANKL in M-cell differentiation[24], the expression of PP M-cell markers, PP size and number were significantly decreased in $Rank^{ΔIE}$ mice, with no obvious difference in the intestine morphology (Supplementary Fig. 3f–h). Consistent with previous work[16,24], loss of M-cells led to a significant decrease in antigen sampling in Peyer's patches along with a concomitant decrease in fecal IgA in $Rank^{ΔIE}$ mice (Supplementary Fig. 3i, Fig. 3e). Importantly, $Rank^{ΔIE}$ mice showed increased susceptibility to DSS-induced colitis (Fig. 3f, g). In addition, the temporal disruption of intestinal epithelial RANK for 2 weeks decreased the expression of M-cell markers in PP and colon LF, and fecal IgA levels in $Rank^{F/F;VilERT2Cre}$ mice. However, no difference in the PP size or number was noted suggesting a possibility for residual NIK signaling in the Peyer's patch at 2 weeks following temporal knockdown of RANK (Fig. 3h–j and Supplementary Fig. 3j, k). Whether the decrease in M-cells following the temporal loss of epithelial RANK-NIK is due to increased apoptosis of M-cells in Peyer's patch needs to be determined. However, ablation of M-cell differentiation in the enteroids from $Nik^{ΔIE}$ strongly suggests that a key role of NIK in M-cell maintenance by regulating the differentiation process. We also show that the temporal loss of M-cells is sufficient to increase the susceptibility to DSS-induced colitis in $Rank^{F/F;VilERT2Cre}$ mice (Fig. 3k, l). Together, the data demonstrate a link between the loss of M-cells and colitis in a model of intestinal epithelial-specific disruption of RANK.

**Loss of epithelial NIK decreases IL17 expression in T cells.** The decrease in gut IgA response in mice with loss of M-cells is not due to a decrease B-cell numbers in the PP (Supplementary Fig. 4a, b). Microarray analysis and qPCR confirmation in the colon of $Nik^{ΔIE}$ mice revealed differential expression of numerous genes involved in B-cell antibody class switching (Supplementary Fig. 4c, d). However, the expression of activation-induced cytidine deaminase (AID) involved in class switching recombination is not changed in the colon of $Nik^{ΔIE}$ mice. To gain further insight, RNA-SEQ analysis was performed in the PP from $Nik^{F/F;VilERT2Cre}$ and $Rank^{F/F;VilERT2Cre}$ mice at 2 weeks after tamoxifen treatment. RNA-SEQ revealed differential expression of 1649 and 851 genes in $Nik^{F/F;VilERT2Cre}$ and $Rank^{F/F;VilERT2Cre}$, respectively (Supplementary Data 1-4). Pathway analysis revealed that RANK-NIK signaling regulates key inflammatory pathways (Supplementary Fig. 4e). Out of 610 genes that were significantly altered in both lines, genes that are associated with M-cells, FAE and B-cells were noted (Fig. 4a). Interestingly, key genes involved in immunomodulation such as *Il17a*, *Il22* and aryl hydrocarbon

receptor (*Ahr*) were significantly decreased in both lines (Fig. 4a). Among these immunomodulatory genes, only *Il17a* expression is associated with luminal sensing of commensals, as revealed by induction of *Il17a* in the PP of germ-free mice following conventionalization (Fig. 4b, c and Supplementary Fig. 4f).

The decrease in *Il17a* mRNA and proteins levels in the PP and colon, respectively, of $Nik^{ΔIE}$, $Ikkα^{ΔIE}$, $Rank^{ΔIE}$, $Nik^{F/F;VilERT2Cre}$, and $Rank^{F/F;VilERT2Cre}$ mice were further confirmed (Fig. 4d and Supplementary Fig. 4g, h). We then questioned how IL17 is decreased by loss of M-cells. M-cell differentiation did not increase *Il17a* mRNA levels in the enteroids (Supplementary Fig. 4i), suggesting that M-cells are not the source of IL17. In the PP, T-cells had the highest levels of *Il17a* expression (Supplementary Fig. 4j). To determine whether the decrease in IL17A is due to a reduction in IL17 expression or a loss in IL17-expressing cell population, flow analysis was performed in $Nik^{F/F;VilERT2Cre}$ chimeras generated using the RORC(γτ)-EGFP mice as donors. No difference in the Th17 cell population (GFP+CD4+ CD3+CD45+; Fig. 4e) suggests that the expression of IL17 was decreased in the PP Th17 cells of $Nik^{F/F;VilERT2Cre}$ mice. No difference was found in the T-regs (FOXP3+CD4+CD3+ CD45+) and T-follicular cells (PD-1+CXCR5+CD4+) in the PP of $Nik^{F/F;VilERT2Cre}$ mice (Fig. 4e). Similarly, no difference in T-cell population was observed in $Rank^{F/F;VilERT2Cre}$ mice (Fig. 4f). Further, flow analysis revealed a significant decrease in IL17+ T-cells in the Peyer's patch of $Nik^{F/F;VilERT2Cre}$ and $Rank^{F/F;VilERT2Cre}$ mice (Fig. 4g, h). We also noted a significant decrease in *Il17a* mRNA expression in flow-sorted CD4+ cells from the PP of $Rank^{F/F;VilERT2Cre}$ (Fig. 4i). We then investigated whether antigenic stimulation (via M-cell) regulates IL17 expression by assessing the PP of global TLR2/4 double knockout ($Tlr2/4^{-/-}$) mice. Expression of *Il17* mRNA was significantly decreased in the PP of $Tlr2/4^{-/-}$ mice (Fig. 4j). Moreover, BMT of WT mice with $Tlr2/4^{-/-}$ donors did not increase *Il17* expression suggesting that TLRs on T-cells are involved in the IL17 response (Fig. 4k)[25]. Together, the data demonstrate that epithelial RANK-NIK signaling is critical for *Il17a* gene expression in the T-cells of intestinal lymphoid follicles.

**Epithelial NIK via IL17 and IgA protects against colitis.** To determine if IL17 could be a potential candidate molecule linking epithelial NIK and gut humoral response, global $Il17^{-/-}$ mice were assessed. Fecal IgA levels were significantly decreased in $Il17^{-/-}$ mice (Fig. 5a); however, PP size, number, and markers of M- or B-cell numbers were not different in the PP of $Il17^{-/-}$ mice (Supplementary Fig. 5a-c). Interestingly, in vitro treatment with recombinant IL17 significantly enhanced IgA production from B-cells (Fig. 5b), suggesting that a cell-autonomous role of IL17 in the production of IgA. However, loss of IL17 decreased the expression of *Igkv1-133* but not *Ighv1-47* and *Igkv9-120* (Supplementary Fig. 5d) demonstrating that IL17 may not directly affect the class switching rather regulate antigen-dependent B-cell differentiation[26]. IL17 is essential for intestinal epithelial barrier function and $Il17^{-/-}$ mice are highly susceptible to DSS-induced colitis (Supplementary Fig. 5e, f)[27,28]. However, local loss of IL17 in the PP and/or colon LF did not affect barrier function in $Nik^{ΔIE}$ mice (Supplementary Fig. 2e, f). It has been shown that B-cell deficient $Jh^{-/-}$ mice that lack IgA are highly susceptible to DSS-induced colitis (Supplementary Fig. 5g-i)[29]. Similarly, IgA-deficient ($IgA^{-/-}$) mice and mice lacking polymeric Ig receptor ($pIgR^{-/-}$) are more susceptible to colitis (Supplementary Fig. 5j, k) demonstrating a critical role of gut IgA in protection against colitis. To determine if dysregulation in IL17 driven gut humoral response promotes colitis in $Nik^{ΔIE}$ mice, $Jh^{-/-}$ mice on 2% DSS for 3-days were transplanted with B-cells isolated from the PP of

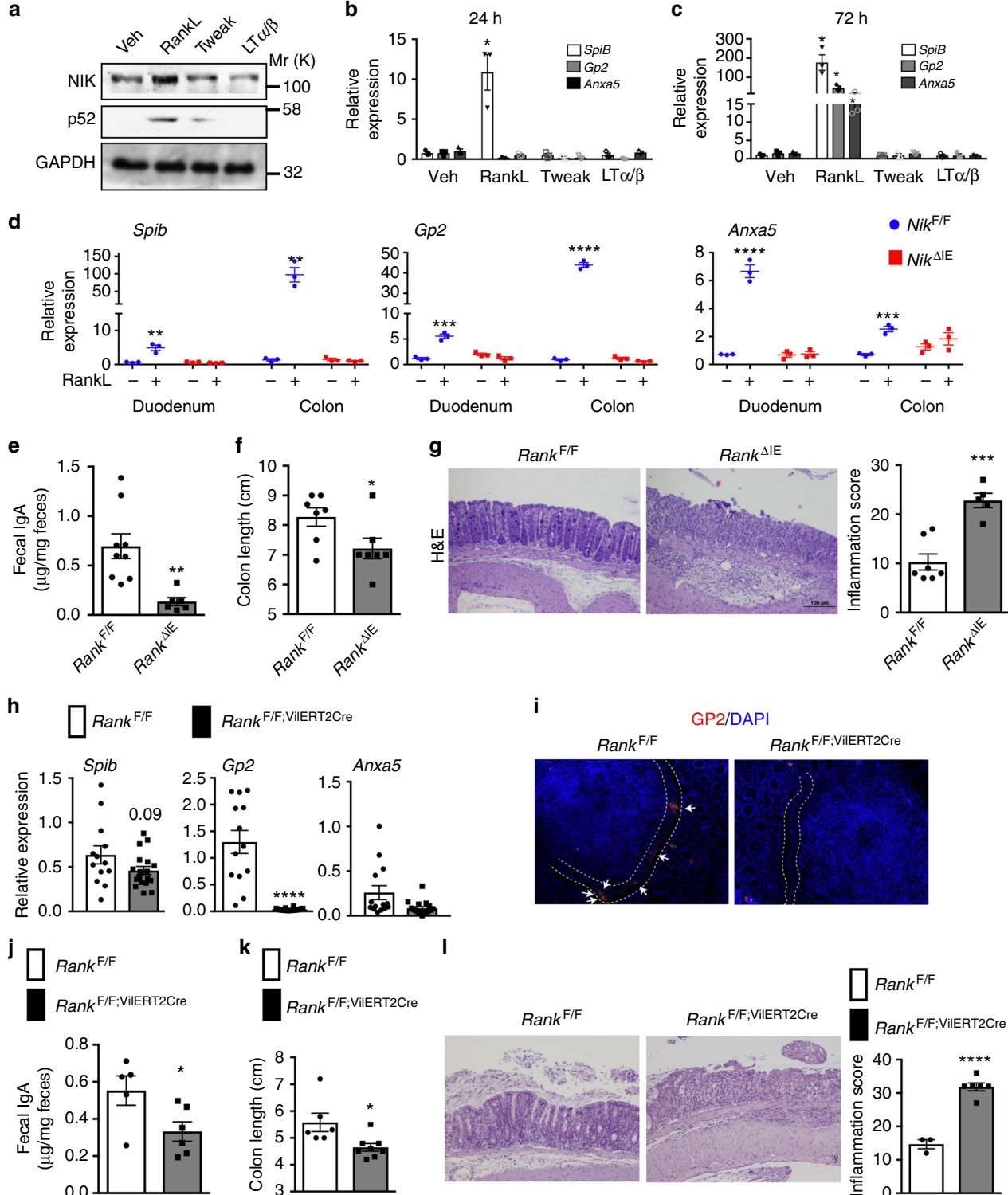

**Fig. 3** Epithelial RANK signaling protects against colitis. **a** Western blot analysis for NIK and p52 in the enteroid cultures treated with vehicle, RANKL, Tweak or LT$\alpha$/$\beta$ for 24 h. **b, c** qPCR analysis of M-cell markers in the duodenal enteroids treated with vehicle, RANKL, Tweak or LT$\alpha$/$\beta$ for 24 h (**b**) or 72 h (**c**). **d** qPCR analysis of M-cell markers in the duodenum and colon enteroids from $Nik^{\Delta IE}$ mice that were treated with RANKL for 72 h. **e** Fecal IgA levels measured in $Rank^{\Delta IE}$ mice. **f, g** Colon length (**f**) and H&E analysis (**g**) assessed in mice with intestinal epithelial-specific disruption of RANK ($Rank^{\Delta IE}$) that were treated with 2% DSS for 6 days. **h** qPCR analysis of M-cell markers in the Peyer's patches of $Rank^{F/F;VilERT2Cre}$ mice at 2 weeks after tamoxifen treatment. **i** GP2 immunostaining in the Peyer's patches of $Rank^{F/F;VilERT2Cre}$ mice at 2 weeks after tamoxifen treatment. Images were taken at 20X magnification. **j** Fecal IgA assessed in $Rank^{F/F;VilERT2Cre}$ mice at 2 weeks after tamoxifen treatment. **k, l** Colon length (**k**) and H&E analysis (**l**) in 8-week-old mice $Rank^{F/F;VilERT2Cre}$ mice treated with DSS after 2 weeks of tamoxifen treatment. Enteroid experiment were done in triplicate and repeated twice. H&E images were taken at ×10 magnification. Results are expressed as mean ± SEM. Significance determined using $t$ test. *$P < 0.05$; **$P < 0.01$; ***$P < 0.001$; ****$P < 0.0001$

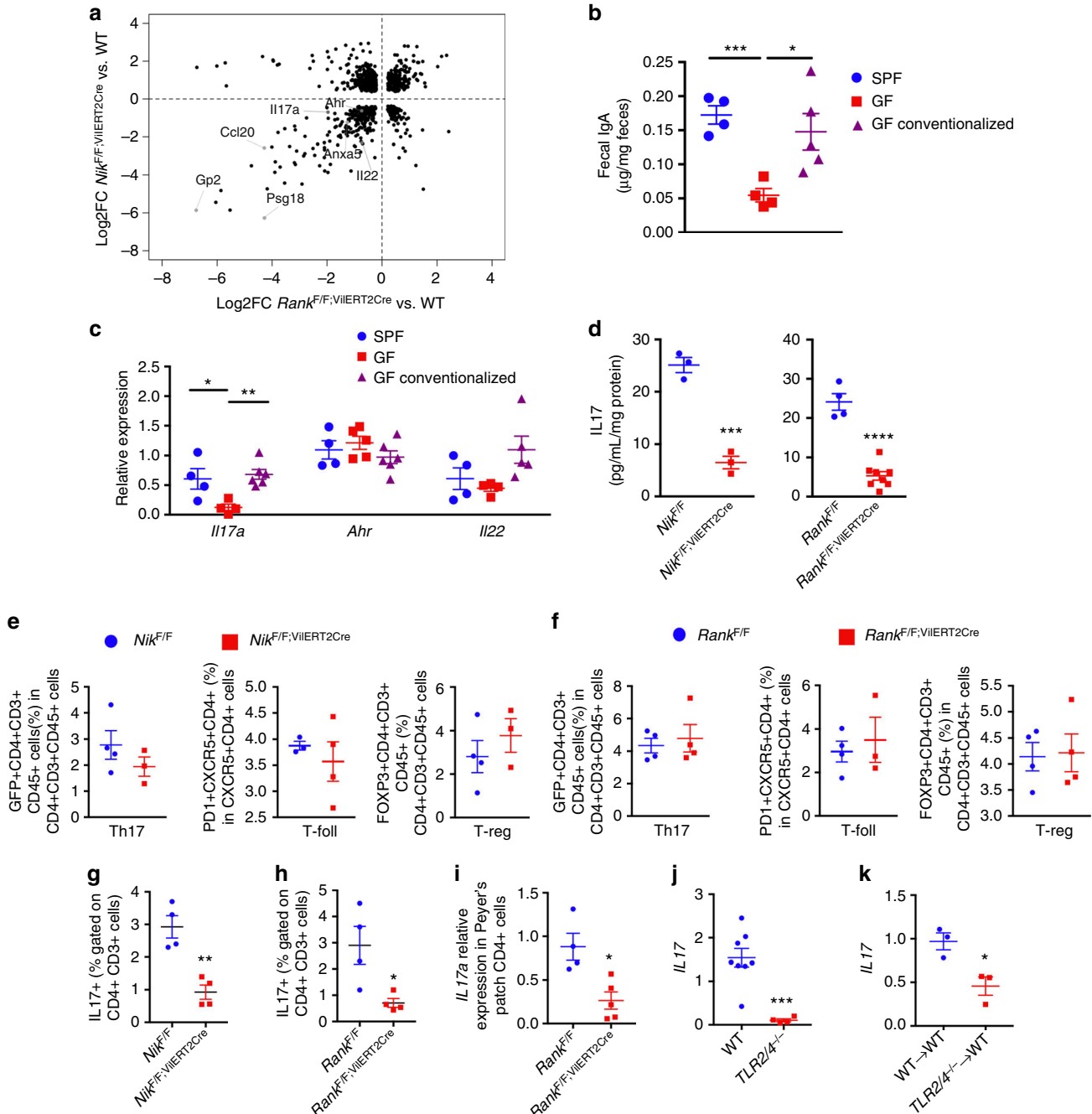

**Fig. 4** Epithelial NIK signaling regulate IL17 expression in T-cells. **a** RNA-seq analysis performed in the Peyer's patches of 7-week-old $Nik^{F/F;VilERT2Cre}$ and $Rank^{F/F;VilERT2Cre}$ mice isolated 2 weeks after tamoxifen treatment. **b, c** Fecal IgA (**b**) and qPCR analysis (**c**) performed in the Peyer's patches of specific pathogen free (SPF), germ free (GF) and GF mice conventionalized for 2 weeks by fecal transplant from SPF donors. **d** IL17 ELISA in the colon of $Nik^{F/F;VilERT2Cre}$ and $Rank^{F/F;VilERT2Cre}$ mice that were treated with 2% DSS for 7 days, following 1 week after disruption of NIK or RANK. **e, f** Flow analysis of T-cells in the PP of $Nik^{F/F;VilERT2Cre}$ (**e**) and $Rank^{F/F;VilERT2Cre}$ mice (**f**). **g, h** Flow analysis to determine the percentage of IL17+ cells in the PP of $Nik^{F/F;VilERT2Cre}$ (**g**) and $Rank^{F/F;VilERT2Cre}$ mice (**h**). **i** qPCR analysis of $Il17a$ mRNA levels in the flow-sorted CD4+ cells from the Peyer's patches of $Rank^{F/F;VilERT2Cre}$ mice. **j** qPCR analysis of $Il17a$ mRNA levels in the Peyer's patches of $Tlr2/4^{-/-}$ mice. **k** qPCR analysis of $Il17a$ mRNA levels in the Peyer's patches of WT mice assessed 2 weeks after transplant with bone marrow from WT or $Tlr2/4^{-/-}$ mice. Results are expressed as mean ± SEM. Significance determined using one-way ANOVA for **b, c** and t-test for **d–i**. *$P < 0.05$; ***$P < 0.001$; ****$P < 0.0001$ vs. $Jh^{-/-}$ WT chimeras

WT, $Nik^{\Delta IE}$, $Rank^{\Delta IE}$, and $Il17^{-/-}$ mice and DSS treatment was continued for additional 4 days (Fig. 5c). B-cell transplant from the PP of $Nik^{\Delta IE}$, $Rank^{\Delta IE}$, and $Il17^{-/-}$ mice increased fecal IgA levels in $Jh^{-/-}$ mice, but to a significantly lower level than the $Jh^{-/-}$ mice that received PP B-cells from WT mice (Fig. 5d). Repletion of IgA significantly improved the colitis index in all the

$Jh^{-/-}$-B-cell chimeras but to a significantly lower level compared to $Jh^{-/-}$-WT B-cell chimera (Fig. 5e–g and Supplementary Fig. 5l). Together, in this very acute model of B-cell transplant where the B-cells maintained the characteristics from the donor mice, epithelial NIK signaling has a protective role against colitis via by regulating intestinal IgA response.

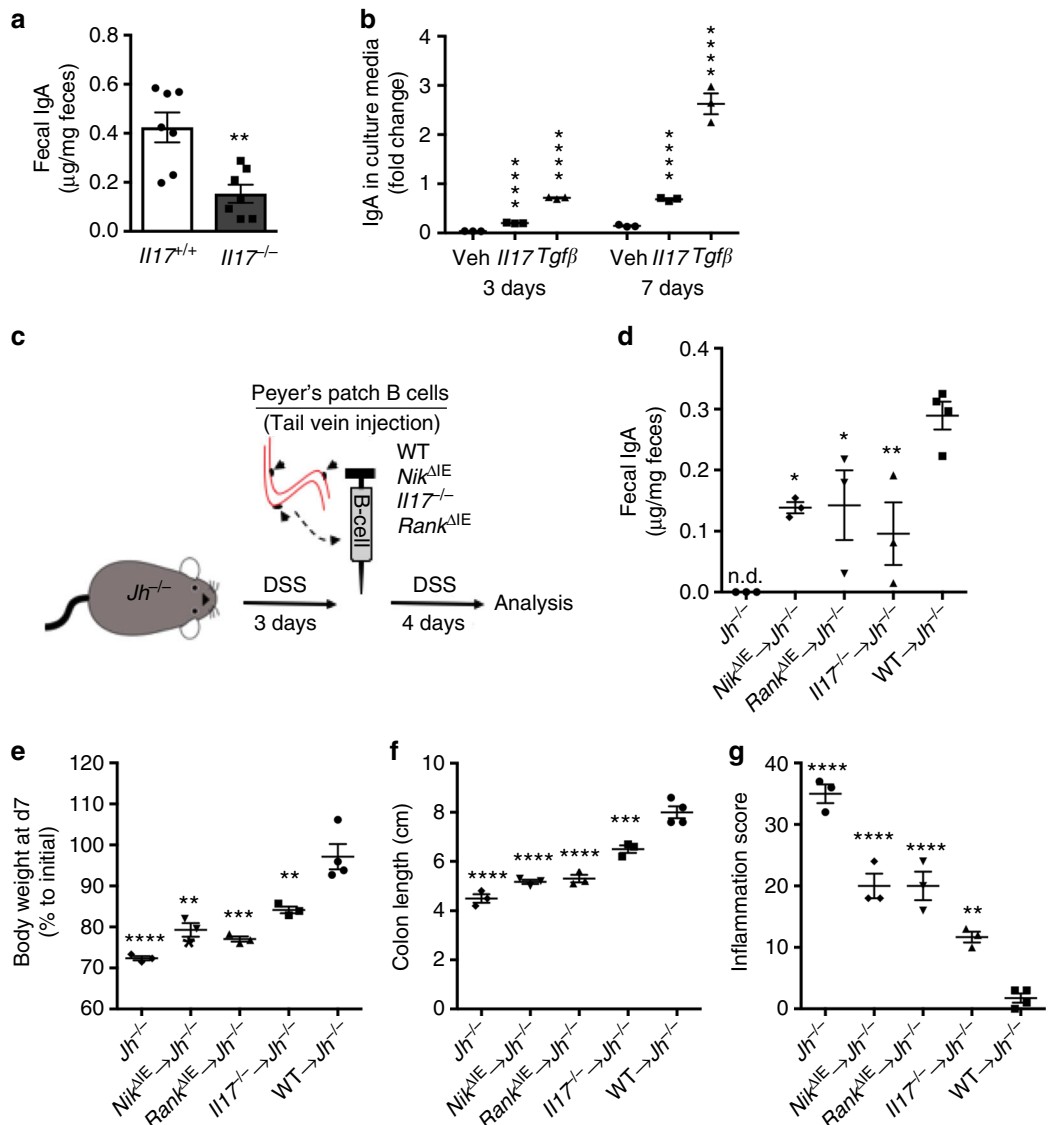

**Fig. 5** IL-17-mediated IgA production is protective in colitis. **a** Fecal IgA in 6-week-old $Il17^{-/-}$ mice. **b** IgA production from PP B-cells at day 3 or 7 following induction with IL17 or Tgfβ. **c** Schematic representation of transplant performed in $Jh^{-/-}$ mice with Peyer's patches B-cells isolated from WT, $Nik^{\Delta IE}$, $Rank^{\Delta IE}$, or $Il17^{-/-}$ mice. **d–g** Fecal IgA (**d**), body weight (**e**), colon length (**f**), and inflammation score (**g**) in 8-week-old $Jh^{-/-}$ chimeras that were treated with DSS for 7 days. Results are expressed as mean ± SEM. Significance determined using $t$-test for **a**, **b** and one-way ANOVA for **d–g**. *$P < 0.05$; **$P < 0.01$; ***$P < 0.001$; ****$P < 0.0001$ vs. $Il17^{+/+}$ or $Jh^{-/-}$ WT chimeras

**Epithelial NIK regulates IgA coating of gut microbiota**. IgA coating of commensal bacteria facilitates immunotolerance by enhancing antigen uptake and processing by immune cells. IgA coating also prevents the colonization of pathogenic or colitogenic bacteria[30,31]. Similar to M-less mice, the decrease in IgA did not alter the microbiota composition in $Nik^{\Delta IE}$ mice (Fig. 6a, b). Moreover, co-housing with $Nik^{\Delta IE}$ mice for 30-days did not increase colitis in the wild-type mice (Supplementary Fig. 6a, b), suggesting that susceptibility to colitis in $Nik^{\Delta IE}$ mice is not con- tributed by microbial diversity. A recent study has shown that mesenchymal specific loss of RANKL decreases bacteria-specific IgA[32]. Here, we demonstrate that temporal loss of epithelial RANK- NIK signaling significantly decreases the levels of IgA-coated bac- teria in the feces of $Nik^{F/F;VilERT2Cre}$ and $Rank^{F/F;VilERT2Cre}$ (Fig. 6c and Supplementary Fig. 6c). Further investigation by qPCR analysis using bacterial-specific primers revealed a significant decrease in the IgA-coated colitogenic bacteria in $Nik^{F/F;VilERT2Cre}$ and $Rank^{F/F;VilERT2Cre}$ mice (Fig. 6d and Supplementary Fig. 6d). To determine

whether the susceptibility to colitis in mice with loss of epithelial NIK is due to an increase in colitogenicity of microbiota, $Nik^{F/F}$ and $Nik^{F/F;VilERT2Cre}$ mice were treated with DSS in the presence or absence of an antibiotic cocktail in the drinking water. DSS sig- nificantly increased colitis in $Nik^{F/F;VilERT2Cre}$, which is partially ameliorated by antibiotic treatment (Fig. 6e, f). This suggests that the maintenance of intestinal immune response by RANK-NIK signaling is essential for IgA coating of the gut microbiota including the colitogenic bacteria.

**Intestinal NIK signaling protects against sepsis**. The role of M- cells in the systemic immune response is not known. IL17 is critical to induce pro-inflammatory cytokines and activation of bactericidal activity in neutrophils[33,34]. Accordingly, in a mouse model of cecal ligation and puncture (CLP), serum IL17 levels were dramatically increased (Supplementary Fig. 7a), and $Il17^{-/-}$ mice show decreased survival following CLP (Fig. 7a). Serum IgA

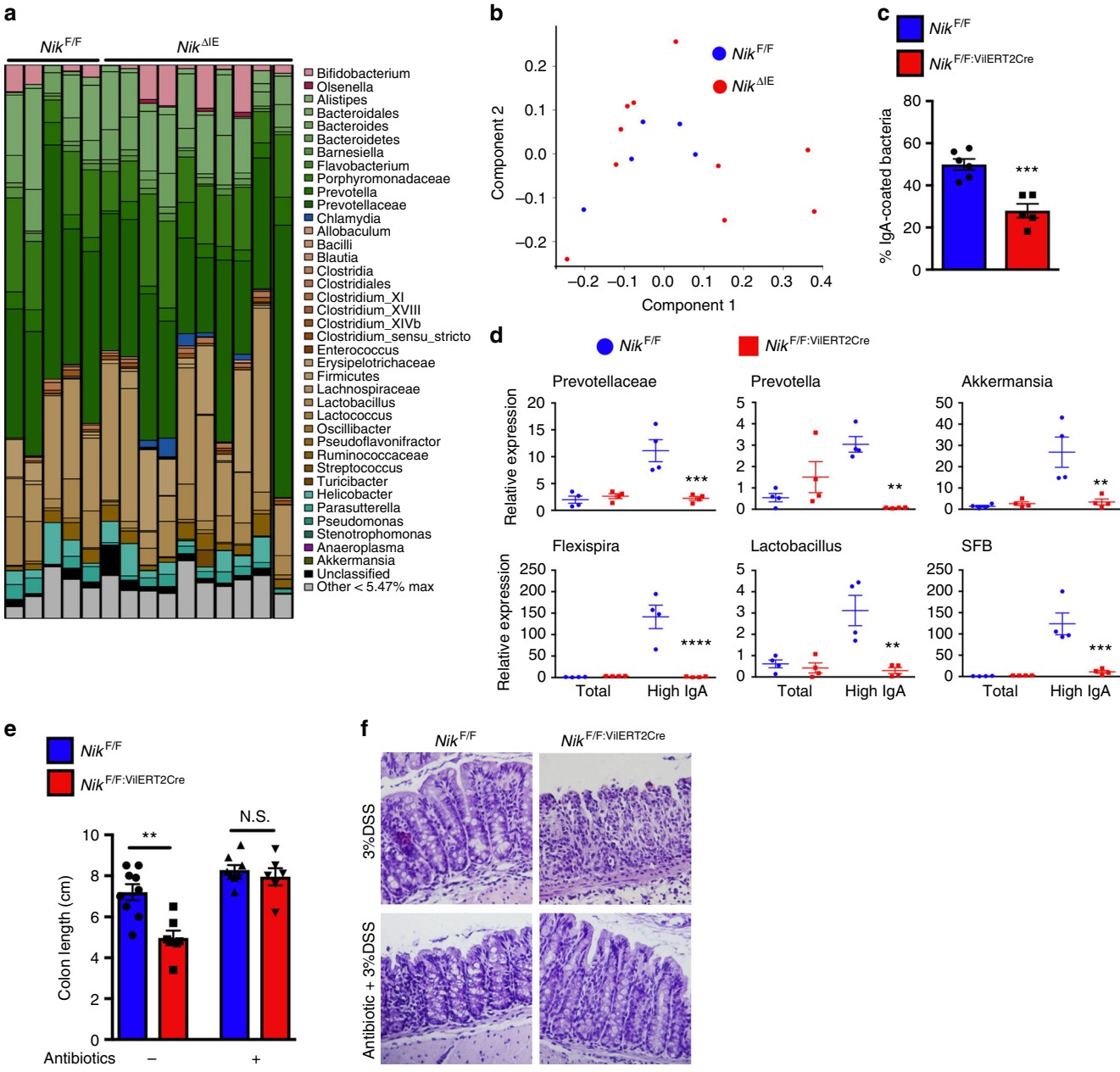

**Fig. 6** Loss of epithelial NIK decreases IgA coating of colitogenic bacteria. **a**, **b** Bacterial composition (**a**) and PC2 plot (**b**) in the feces of 6-week-old $Nik^{F/F}$ and $Nik^{\Delta IE}$ mice assessed by 16S genomic DNA sequencing. **c** Percentage of IgA-coated bacteria in the feces of $Nik^{F/F;VilERT2Cre}$ mice assessed by flow analysis. **d** qPCR analysis for bacterial DNA assessed in flow-sorted IgA-coated bacteria in the feces of $Nik^{F/F;VilERT2Cre}$ mice. Significance determined using $t$ test. **$P < 0.01$; ***$P < 0.001$; ****$P < 0.0001$. **e**, **f** Colon length (**e**) and H&E analysis (**f**) in DSS-treated $Nik^{F/F;VilERT2Cre}$ mice in the presence or absence of antibiotic cocktail. H&E images were taken at ×10 magnification. Results are expressed as mean ± SEM. Significance determined using one-way ANOVA. **$P < 0.01$

has a protective role in the clearance of bacteria that escapes from the intestine[35]. Interestingly, the circulating levels of IgA were significantly lower in $Il17^{-/-}$ mice (Fig. 7b). To understand whether circulating IL17 or IgA protects against sepsis, CLP was performed in $IgA^{-/-}$ mice that lacked both serum and fecal IgA (Supplementary Fig. 7b, c). The mortality was significantly increased in $IgA^{-/-}$ mice after CLP, independent of increases in serum IL17 levels (Fig. 7c and Supplementary Fig. 7d). IgA coating of the microbiota regulates bacterial pathogenicity. Therefore, to determine if maintaining circulating IgA will protect against sepsis even in the absence of luminal IgA, $pIgR^{-/-}$ mice, which have serum IgA but not fecal IgA, were used (Supplementary Fig. 7e, f). We did not observe any difference in the

serum IL17 levels and the survival rate between $pIgR^{+/+}$ and $pIgR^{-/-}$ mice following CLP (Fig. 7d and Supplementary Fig. 7g), suggesting that circulating IgA plays a key role in the protection against sepsis.

To determine the systemic effect of decreased serum IgA, CLP was performed in $Rank^{F/F;VilERT2Cre}$ and $Nik^{F/F;VilERT2Cre}$ mice. Interestingly, the induction of serum IL17 following CLP was significantly abrogated in $Rank^{F/F;VilERT2Cre}$ and $Nik^{F/F;VilERT2Cre}$ mice (Fig. 7g, h). Moreover, loss of intestinal RANK and NIK significantly decreased the survival following CLP (Fig. 7i). Together, the data demonstrate a far-reaching role of intestinal-epithelial NIK signaling in the systemic inflammatory response and polymicrobial sepsis.

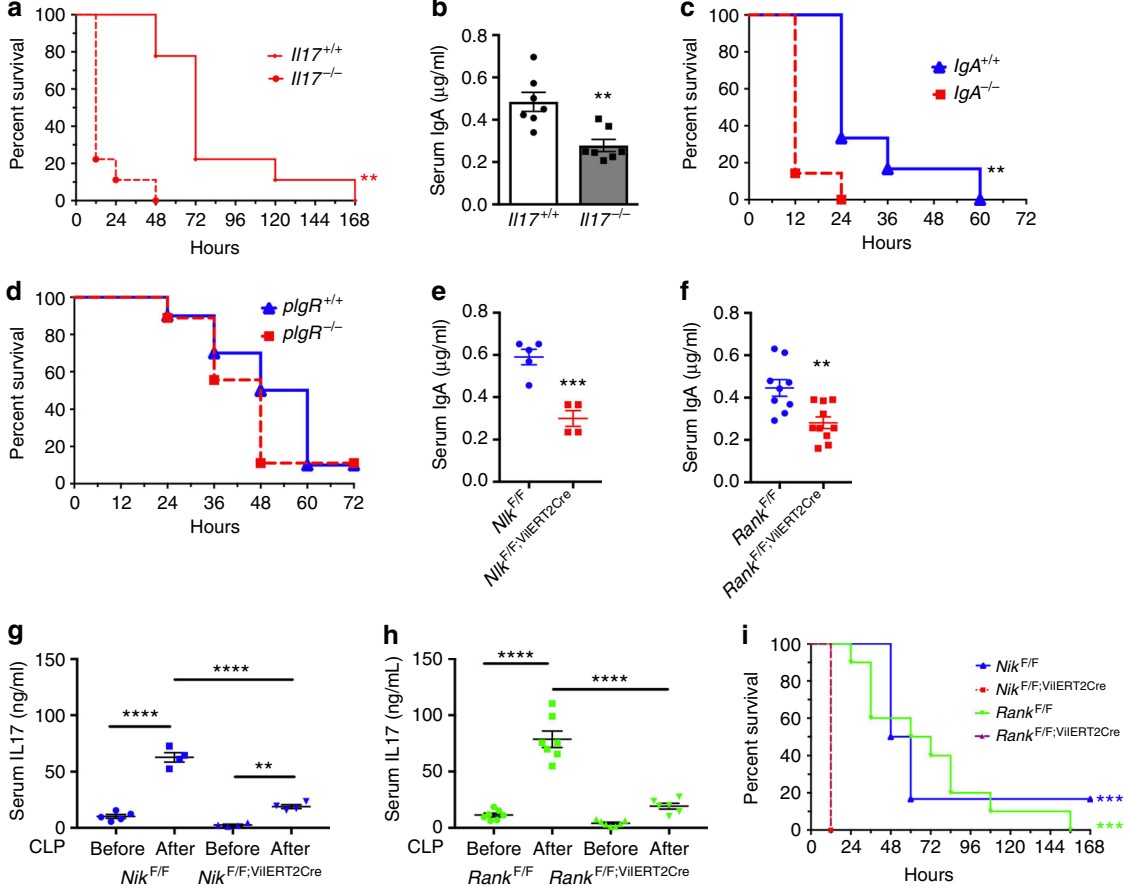

**Fig. 7** Loss of epithelial RANK/NIK accentuates polymicrobial sepsis. **a** Percent survival assessed in 6-week-old $Il17^{-/-}$ mice following cecal ligation and puncture (CLP); $n = 9$ mice/group. **b** Serum IgA assessed in 6-week-old $Il17^{-/-}$ mice. **c** Percent survival assessed in 6-week $IgA^{-/-}$ after CLP; $n = 6$-7 mice/group. **d** Percent survival assessed in 6-week $pIGR^{-/-}$ after CLP. **e, f** Serum IgA in $Nik^{F/F;VilERT2Cre}$ (**e**), and $Rank^{F/F;VilERT2Cre}$ mice (**f**). **g–i** Serum IL17 in $Nik^{F/F;VilERT2Cre}$ (**g**) and $Rank^{F/F;VilERT2Cre}$ mice (**h**) assessed at 6 h post CLP, and percent survival assessed after CLP (**i**). For CLP $n = 6$-9 mice/group. Results are expressed as mean ± SEM. Significance determined using $t$ test or one-way ANOVA or log-rank test (survival). $*P < 0.05$; $**P < 0.01$; $***P < 0.001$; $****P < 0.0001$

**Constitutive activation of NIK signaling exacerbates colitis.** The inflammatory response to antigenic challenges is tightly regulated in the intestine, and chronic activation of these pathways increases intestinal injury[6,36–40]. Our data show that disruption of intestinal NIK signaling increases the susceptibility to colitis. However, in ulcerative colitis patients, a robust increase epithelial NIK signaling was observed (Supplementary Fig. 8a, b). Therefore, we hypothesized that non-canonical NFκB signaling may act as a homeostatic signaling pathway, where either disruption or activation could lead to enhanced susceptibility to intestinal injury. To test our hypothesis, we generated mice with intestinal epithelial-specific overexpression of NIK using a mutant of NIK (resistant to degradation) cloned in the ROSA locus downstream of a LoxP-stop-LoxP cassette[41] ($Nik^{VilRosaΔT3}$; Fig. 8a). A robust activation of intestinal epithelial NIK signaling (Fig. 8b), did not cause any overt inflammatory damage, however, several cytokines were significantly elevated in the colon (Supplementary Fig. 8c, d). Interestingly, overexpression of NIK increased the expression of M-cell markers and GP2 staining in the normal colon epithelium (Fig. 8c, d), but not in the small intestinal epithelium or PP or colon LF (Supplementary Fig. 8e–g). IL17 levels were significantly elevated in the colon and serum of $Nik^{VilRosaΔT3}$ mice (Fig. 8e); however, no difference in fecal IgA was noted suggesting that IL17 does not further augment IgA production in $Nik^{VilRosaΔT3}$ mice (Supplementary Fig. 8h).

$Nik^{VilRosaΔT3}$ mice were highly susceptible to DSS and also to *Salmonella*-induced colitis, with signs of severe intestinal bleeding, diarrhea, and infiltration of immune cells in the colon (Fig. 8f–h and Supplementary Fig. 8i, j). Together, the data indicate that constitutive activation of epithelial NIK induces a pro-inflammatory response and increases the susceptibility to colitis.

## Discussion

Intestinal homeostasis is maintained by a unique basal inflammatory tone that is regulated by numerous pathways. Dysregulation of these pathways is associated with increased intestinal injury[6,36–40]. In this study, we demonstrate that epithelial non-canonical NFκB signaling is an essential homeostatic pathway in the regulation of intestinal inflammatory response. In the intestine, epithelial NIK signaling is essential for M-cell maintenance and protection against colitis. The decrease in the expression of IL17 on T-cells and gut humoral response in mice with epithelial loss of RANK-NIK signaling correlates with a decrease in the antigen sampling M-cells. In mouse models and patients with colitis, epithelial non-canonical NFκB is highly active. Our novel mouse model with constitutive expression of intestinal epithelial NIK exhibit exacerbated colitis which is associated with an increase in IL17 and ectopic M-cells in the colon (Fig. 8i). Thus,

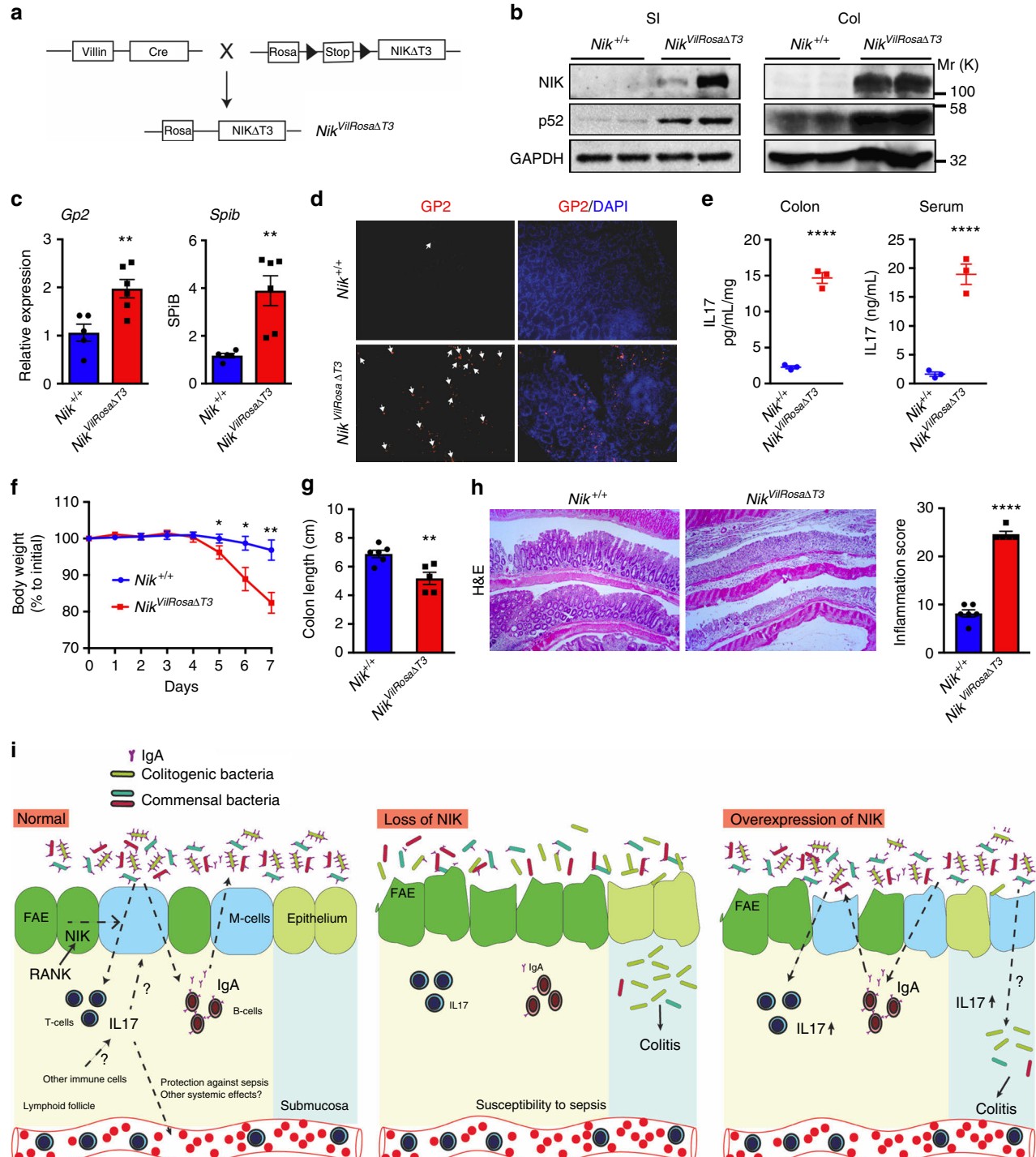

**Fig. 8** Constitutive activation of NIK induces ectopic colonic M-cells and predisposes to colitis. **a** Schematic representation of the mice with intestinal epithelial-specific overexpression of NIK (*Nik*^VilRosaΔT3^) generated by crossing mice with cDNA harboring mutation in TRAF3 binding site of NIK driven by loxP-Rosa26 promoter-loxP with mice expressing Cre recombinase driven by Villin promoter. **b** Western blot analysis of NIK and p52 in the small intestine (SI) and colon (Col) of 6-week-old *Nik*^VilRosaΔT3^ mice. **c, d** qPCR analysis of M-cell markers (**c**) and GP2 staining (**d**) in the colon of *Nik*^VilRosaΔT3^ mice. Images were taken at ×10 magnification. **e** ELISA for IL17 in the colon and serum of 6-week-old untreated *Nik*^VilRosaΔT3^ mice. **f–h** Body weight (**f**), colon length (**g**), and H&E analysis (**h**) in 7-week-old *Nik*^VilRosaΔT3^ mice that were treated with DSS for 6 days. H&E images were taken at ×10 magnification. Results are expressed as mean ± SEM. Significance was determined using *t* test. *P < 0.05; **P < 0.01; ***P < 0.001. **i** Intestinal epithelial NIK-RANK signaling is critical to maintain M-cells and local and systemic IL17 and IgA response. Intestinal epithelial non-canonical NFκB signaling is essential to maintain the antigen sampling M-cells in the Peyer's patch. Loss of M-cells in the Peyer's patches is associated with a decrease in IL17 expression in T-cells. Loss of epithelial NIK signaling diminishes both local and systemic IL17 and IgA response resulting in increased susceptibility to colitis and polymicrobial sepsis. Conversely, constitutive activation of epithelial NIK also exacerbates colitis, which is associated with induction in ectopic colonic M-cells and IL-17 levels. Thus, intestinal NIK signaling plays a rheostatic role in maintaining intestinal homeostasis by regulating the immune and inflammatory signatures

intestinal epithelial NIK signaling acts as a rheostat in maintaining intestinal homeostasis.

The canonical NF-κB1 pathway is well documented as a critical signaling pathway in intestinal inflammation. Intestinal epithelial-specific conditional knockout mice demonstrated that canonical NF-κB1 pathway is essential for epithelial cell survival and protection against colitis[4,6,39]. Interestingly, constitutive activation of NF-κB1 pathway in intestinal epithelial cells could also lead to enhanced pro-inflammatory mediator expression and injury[5]. Unlike the canonical NF-κB1 pathway, the role of intestinal epithelial non-canonical NF-κB2 was unclear[9–12]. Our data show a clear rheostatic role of NF-κB2 in intestinal inflammation. Although the phenotypes are similar with respects to disruption or overexpression of the canonical and non-canonical arms of NF-κB in intestinal epithelial cells, the underlying mechanisms remain distinct. The epithelial non-canonical NF-κB2 does not directly alter classical NF-κB pro-inflammatory targets or cell survival, but it is essential in maintaining M-cells. Data from both constitutive and temporal models of NIK, RANK, and IKKα mice, all part of the non-canonical pathway, share several features such as decreased M-cells, IL17, and IgA. We suggest a novel axis in colitis where loss of intestinal epithelial NIK signaling decreased M-cells on the Peyer's patch, which is associated with dysregulated local and systemic levels of IL17 and IgA. Further investigation with M-cell-specific Cre transgenic mice to knock-out NIK signaling will help us to exclude the confounding effect of non-canonical NF-κB2 in other epithelial cell types in the colitis phenotype. Moreover, we cannot completely exclude NIK-dependent mechanisms that are independent of the non-canonical NF-κB2 pathway. Subunits of the NF-κB pathway have functions that are independent of its role in the pathway. Intestinal epithelial IKKα, a subunit shared by both arms of the NF-κB pathway, is an important regulator of ILC3 cells and autophagy[42,43]. Future studies are needed to clearly understand M-cell autonomous role of non-canonical NF-κB2 and how both arms of the NF-κB pathway regulate specific functions for each subunit and how the integration of these pathways could impact intestinal inflammation.

It has long been inferred that M-cells are integral in maintaining the inflammatory tone of the intestine due to its role in bacterial and IgA transport[16,44]. Here, using mouse models of colitis, we demonstrate that loss of intestinal-epithelial RANK and NIK is associated with a decrease in the expression of pro-inflammatory IL17A in the intestinal LF. Recent studies in $Il17^{-/-}$ mice have raised questions about the exclusive pro-inflammatory role of IL17. Specifically, $Il17^{-/-}$ mice are highly susceptible to colitis due to intestinal epithelial barrier dysfunction[27,33,34,45]. In $Nik^{\Delta IE}$, local loss of IL17 in the Peyer's patch or colon lymphoid follicles did not affect barrier function or induce pro-inflammatory mediators at steady state. Our work demonstrates an additional mechanism by which IL17 protects against colitis. Conventionalization of microbiota increased the expression of IL17 in the Peyer's patches and IL17 has been shown to induce IgA secretion in a cell-autonomous manner[26]. Therefore, the decrease in fecal IgA in mice with loss of intestinal epithelial NIK signaling may be due to partial loss of IL17-dependent B-cell humoral response. The expression of IL-22 and Ahr were also significantly decreased in the Peyer's patches of $Nik^{\Delta IE}$ mice. IL-22 improves intestinal barrier function by regulating the expression of epithelial tight junction proteins and Ahr protects against colitis by suppressing T-cell response[46–48]. Further studies are required to understand the IL17-dependent and independent function of IL-22 and/or Ahr signaling in the protective role of M-cells in colitis. Studies with $Il17r^{-/-}$ mice show the preferential establishment of colitogenic bacteria without changes in commensal microbial population[49]. Recently,

Palm et al., demonstrated that a decrease in the high-IgA coating of the colitogenic bacteria exacerbates DSS-induced colitis[30]. In support of the notion, oral administration of high-affinity IgA shifts the balance towards the enrichment of gut-friendly commensals[50]. IgA coating of the bacteria correlates with the severity of inflammation and pathogens coated with IgA are efficiently neutralized by the immune system. Here, we demonstrate that temporal loss of epithelial RANK-NIK signaling is sufficient to dysregulate IgA production including IgA against colitogenic bacteria. Therefore, we speculate that the decreased IgA coating of the colitogenic bacteria may have resulted in increased pathogenicity resulting in severe colitis in mice with intestinal-epithelial-specific disruption of RANK-NIK signaling. Further investigation in mice with M-cell specific loss of RANK-NIK signaling is required to establish the critical role of continuous antigen sampling via M-cells in the maintenance of gut IgA response even in the presence of memory B-cells.

At present, the mechanisms by which inflammatory cytokines and infectious models (such as *Salmonella* and *C. rodentium*) induce M-cells are not well-understood[51–53]. Overexpression of p52-RelB failed to induce M-cells and similarly, constitutive epithelial NIK signaling was not sufficient to increase M-cells in PP or colon LF. However, epithelial NIK overexpression induced ectopic expression of M-cell markers in the colon and exacerbates colitis. Whether the increase in the steady-state levels of IL17 is due to enhanced bacterial entry via the ectopic colonic M-cells requires further investigation. Although it appears paradoxical where knockout and activation of the same pathway leads to intestinal injury, our observations are in line with many pathways such as the classical NFκB pathway, inflammasomes, and STAT3 pathway where either disruption or chronic activation may lead to enhanced injury[6,36–40].

One of the more surprising findings is the systemic role of M-cell axis in polymicrobial sepsis. IL17 has a complex role in sepsis. Appropriate levels of IL17 are required for a protective response and dysregulated levels can lead to a decrease in survival in mouse models of sepsis[34,54,55]. Our study demonstrates a critical role of intestinal epithelial RANK-NIK signaling in the systemic regulation of IL17 following polymicrobial sepsis. Increased mortality in $IgA^{-/-}$ mice indicates that IL17 alone is not sufficient to protect against polymicrobial sepsis. However, systemic IL17 may play a role in sepsis in-part by inducing IgA production from the circulating or resident B-cells, and also by regulating tissue-specific inflammation[45,49]. It should be noted that low serum IgA levels increase the mortality rate due to septic shock in transplant patients[56]. Therefore, understanding the contribution of intestinal M-cells in systemic immune response will be helpful to develop new therapeutic strategies in the treatment of polymicrobial sepsis.

Although the prognostic use of high-affinity IgA levels in colitis needs further investigation, the decrease in inflammation in mice following administration of high-affinity IgA provides an alternative treatment for colitis[50]. In humans, common variable immunodeficiency (CVID), an immunodeficiency syndrome, is often associated with secondary infections. About 10–20% of CVID patients develop ulcerative colitis through mechanisms that are unclear[57]. Recent studies have identified a heterozygous frameshift mutation in NFκB2 in CVID patients[58]. Similarly, biallelic loss of function mutation in NIK results in B-cell lymphopenia and decreased B-cell differentiation due to an impairment in the processing of NFκB2 to p52[8,59–61]. Therefore, it will be interesting to investigate whether the increased susceptibility to colitis in CVID patients is due to dysregulation in the epithelial NIK-M-cell-IgA response.

In conclusion, we show that intestinal epithelial non-canonical NFκB signaling acts as a critical homeostatic regulator of

intestinal homeostasis. This work demonstrates a novel mechanistic finding where epithelial NIK signaling plays a protective role in colitis and polymicrobial sepsis by shaping intestinal and systemic immune responses.

## Methods

**Animal studies**. $Nik^{F/F}$, $IKK\alpha^{F/F,62}$, $Rank^{F/F,16}$, and $Nik^{loxP\text{-}Rosa26\text{-}loxP,41}$ mice were crossed with mice that express Cre recombinase under the control of villin promoter to generate $Nik^{\Delta IE}$, $IKK\alpha^{\Delta IE}$, $Rank^{\Delta IE}$, and $Nik^{VilRosa\Delta T3}$ mice, respectively. $Nik^{F/F;VilERT2Cre}$ and $Rank^{F/F;VilERT2Cre}$ were generated by crossing $Nik^{F/F}$ and $Rank^{F/F}$ with mice that express tamoxifen-inducible Cre-recombinase. $Nik^{-/-}$, $Il17^{-/-}$, $Jh^{-/-}$, and RORC($\gamma\tau$)-EGFP, $Tlr2/4^{-/-}$, $IgA^{-/-}$, and $pIgR^{-/-}$ mice were described earlier[27,29,63–66]. All mice used in this study were on a C57BL/6 background. All mice were fed ad libitum with standard chow diet (Research diets, New Brunswick, NJ) and kept in a 12-hour dark/light cycle. All animal studies were carried out in accordance with Association for Assessment and Accreditation of Laboratory Animal Care International guidelines and approved by the University Committee on the Use and Care of Animals at the University of Michigan.

**Ulcerative colitis samples**. Deidentified biopsies from IBD patients were collected under the an IRB: HUM00041845, approved at the University of Michigan.

**Animal treatments**. For DSS treatment, animals were provided with drinking water containing 2% DSS for 5-7 days, as detailed in the figure legends. In $Nik^{F/F;VilERT2Cre}$ and $Rank^{F/F;VilERT2Cre}$ mice, DSS was started at 10 days after tamoxifen treatment (100 mg/kg body weight, intraperitoneal). For permeability assays, $Nik^{F/F}$ and $Nik^{\Delta IE}$ mice were treated with 2% DSS for 6 days and then gavaged with FITC-dextran (0.5 mg/g body weight) and serum was collected 4 h post treatment. Serum FITC concentration was measured at 485 nm excitation and 528 nm emission wavelength and the results were expressed as arbitrary relative fluorescent units. For antibiotics treatment, animals were given water containing antibiotic cocktail (Ampicillin 1 g/L, neomycin 1 g/L, gentamycin 500 mg/L; penicillin 100U/L) in the drinking water, ad libitum. In addition, oral gavage of vancomycin (1 mg/mL) and metronidazole (0.5 mg/mL) was given on alternate days for 30 days and then treated with 2% DSS or 2% DSS with antibiotics in water for 7 days. For RNA and protein extraction, either individual or pooled PP or colon LF were analyzed. Similarly, small intestine or colon epithelial scrapes were analyzed for RNA and protein. For H&E staining, intestinal tissues were formalin fixed immediately after dissection and paraffin embedded after 24 h. Six micron sections were stained and the inflammation index was scored by an independent pathologist in a blinded manner[67].

**Isolation of Peyer's patches and colon lymphoid follicles**. Peyer's patches were dissected out from the intestine for RNA, protein and immunostaining. For colon lymphoid follicles, colon was flushed with sterile PBS and LF were isolated under a dissection microscope. Two to four lymphoid follicles could be isolated from the colon of each mice.

**Antigen sampling using microbeads**. Antigen sampling using microbeads was performed in $Nik^{F/F}$ and $Nik^{\Delta IE}$ mice that were fasted for 4 h and then gavaged with Flouresbrite YG microspheres (0.2 μM), Polysciences Inc. Warrington, PA[68]. Twenty-four hours following gavage, animals were euthanized, Peyer's patches were frozen using OCT. Frozen sections were cut using cryostat, counterstained using DAPI and the uptake of microspheres was visualized using Leica fluorescent microscope.

**Bone marrow and PP B cell transplant**. For bone marrow transplant, mice were irradiated with 6gy radiation twice at 4 h interval and then tail vein injected with $2\times10^6$ bone marrow hematopoietic stem-cells from the donors. For B-cell transplant, PP from 10 mice was pooled and B-cells were enriched (Miltenyi Biotec). $Jh^{-/-}$ were placed on 2% DSS for 3 days and then injected with $1\times10^6$ B-cells intravenously and were continued on DSS for additional 4 days.

**Real-time quantitative PCR**. 1 μg of total RNA extracted using trizol reagent (Qiagen) from individual PP or colon LF were reverse transcribed and gene expression was analyzed by quantitative PCR (qPCR) with SYBR green master mix (Radiant). All genes were normalized to β-actin and the results are expressed as relative fold change. The primers used in the study are listed in Supplementary Table 1. For RNA analysis, at least 2–3 Peyer's patches were analyzed per mice and the $N$ number in the figure legends denotes the number of mice used per group.

**16S rRNA gene sequencing and bacterial community analysis**. Bacterial sequencing analysis was done at the Microbial Systems Molecular Biology Lab, a part of the University of Michigan Host Microbiome Initiative. Briefly, the V4 region of the 16srRNA gene was amplified from each sample using the Dual-indexing sequencing strategy[69]. PCR was performed using the following

conditions: 95 ° C, 2 min; [95 °C, 20 s; 55 °C, 15 s] – 30 cycles; 72 °C, 10 min; 4 °C using Accuprime High Fidelity Taq (ThermoFisher, Grand Island, NY). Amplicons were visualized using the eGel 96, 2% SYBR Safe gel system (ThermoFisher), and samples were normalized using the SequalPrep Normalization plate kit, 96-well (ThermoFisher). Sequencing was done on the Illumina MiSeq platform, using a MiSeq Reagent Kit V2 500 cycles, according to the manufacturer's instructions with modifications[69]. Bacterial community analysis was done based on Mothur wiki.

**RNAseq data analysis**. RNA was isolated from the Peyer's patches of five to eight $Nik^{F/F;VilERT2Cre}$ and $Rank^{F/F;VilERT2Cre}$ mice and their WT littermates, and then RNA-seq and computational analysis were performed using the flux high-performance computer cluster hosted by Advanced Research Computing (ARC) at the University of Michigan. Sequencing read quality was assessed utilizing FastQC. A splice junction aware build of the mouse genome (mm10) was built using the genomeGenerate function from STAR 2.5.2[70]. Read pairs were aligned to the genome using STAR, using the options "outFilterMultimapNmax 10" and "sjdbScore 2". Quantification and differential expression analysis between WT, $Nik^{F/F;VilERT2Cre}$ and $Rank^{F/F;VilERT2Cre}$ mice were conducted using CuffDiff v 2.1.1 with the parameter settings "-compatible-hits-norm," and "–frag-bias-correct" . UCSC mm10.fa and the GENCODE mouse M12 primary assembly annotation GTF were used as the reference genome and reference transcriptome, respectively. Genes were considered differentially expressed at a false-discovery rate-adjusted $P$ value of <0.05.

**Immunostaining**. For immunostaining, 6-micron sections were performed with formalin fixed or OCT-embedded frozen tissues and then fixed with PBS-buffered formalin for 15 min and permeabilized using 0.05% Triton X-100 for 10 min. Sections were blocked in 10% goat serum in PBST for 30 min at room temperature, and probed with polyclonal rabbit anti-Ki67 (Catalog # VP-RM04, Vector Laboratories, Burlingame, CA), Anti-GP2 (Catalog # D278-5, mAb-PE; MBL International, Woburn, MA), NIK (Catalog # sc-7211, Santa Cruz Biotechnology Inc, Dallas, TX) overnight at 4 °C. Images were taken under ×20 magnification.

**Western blot analysis**. Whole-cell extract from the PP, colon LF or intestinal epithelial cell scrapes were prepared using RIPA buffer (50 mM Tris HCl pH 7.5, 150 mM NaCl, 2 mM EDTA, 1% NP40, 0.1% SDS) with protease and phosphatase inhibitor. 30 μg of protein loaded per well and immunoblotted overnight at 4 °C with antibody against NIK (Catalog # 4994, Cell signaling technology, Danvers, MA), p52 (Catalog # 4882, Cell signaling technology), E-cadherin (Catalog # 3195, Cell signaling technology), Occludin (Catalog # 13409-1-AP, Proteintech, Rosemont, IL), Actin (Catalog # 66009-1-Ig, Proteintech Inc) and GAPDH (Catalog # sc-25778, Santa Cruz Biotechnology Inc). All the primary antibodies were used at a dilution of 1:1000. HRP-conjugated or licor secondary antibodies (Catalog #7074, #7076, #5470, #5151, Cell signaling technology) and immunoblots developed using chemidoc touch imaging system (ChemiDoc, BioRad, Herculus, CA) or Odyssey Infrared Clx imaging system, Licor Biosciences.

**Enteroid culture and treatment**. For generating enteroids, the small intestine or colon were cut longitudinally and incubated for 15 min at room temperature in DPBS containing 2.5 μg/mL amphotericin B, 25 μg/mL gentamicin and 50 μg/mL normocin. The intestines were incubated in 10 mM DTT for 15 min at room temperature and followed by gentle rotation in 8 mM EDTA at 4 °C for 75 min. After three washes with DPBS, the crypts were separated by snap shaking. The crypts were washed with DPBS containing antibiotics and then pelleted by spinning at $40 \times g$ for 2 min at 4 °C. The pellet was suspended in a solution of 66% Matrigel (Corning, Corning, NY), 33% LWRN medium, and 10 μM Rock inhibitor (Y27632; Miltenyi, San Diego, CA) at a concentration of 2 crypts/μL. A volume of 250 μL of crypt suspension were plated in a 6-well plate. On the fourth day of culture, enteroids were treated with either vehicle or 100 ng/mL human recombinant RANKL (ProSpec, East Brunswick, NJ), 100 ng/mL Tweak (PeproTech Inc., Rocky Hill, NJ), 10 ng/mL Ltα/β for 24- or 72-h and then processed for RNA or protein or for histological analysis. For UEA staining, frozen enteroid sections were stained with FITC-conjugated-UEA1 for 1-h in dark at room temperature and then washed with PBST thrice before imaging.

**CLP sepsis model**. The abdomen of the anesthetized mice was shaved and disinfected using betadine solution, followed by wiping with a 70% alcohol swab. Under aseptic conditions, a 1–2 cm midline laparotomy was performed to expose the cecum with adjoining intestine. The cecum was tightly ligated with a 5-0 suture (Nylon suture, 697; Ethicon) at its base below the ileocecal valve, and perforation was done twice (at >1 cm from the cecal end) with a 23-gauge needle on the same side of the cecum. Cecal contents were gently extruded and cecum was returned to the peritoneal cavity. The peritoneum and the skin were closed with 6.0 nylon sutures and 9mm Autoclips (205016, MikRon Precision Inc), respectively. Mice were resuscitated by injecting subcutaneously 500 μl of pre-warmed saline solution using a 27G needle. The animals were placed on a work surface until recovery. Mice were monitored every 12 h for survival. For control experimental animals, sham laparotomy was performed without ligation and puncture.

**ELISA for colon IL17**. For IL17 ELISA in the colon, mice were treated with 2% DSS for 6–7 days and the whole colon was collected. IL17 was measured in the colon lysates or serum using mouse IL17 ELISA kit following manufacturer's recommendations (Lifespan Biosciences Inc, Seattle, WA). IL17 in colon tissues were normalized to the protein levels.

**Fecal and serum IgA by ELISA**. Fecal and serum IgA were measured using mouse-specific IgA kit following manufacturers protocol (Bethyl Laboratories, Montgomery, TX). Briefly, feces was resuspended to 10% fecal slurry in sample dilution buffer. The fecal slurry was then diluted 1:200 in sample dilution buffer (50 mM Tris, 0.14 mM NaCl, 1% BSA and 0.05% Tween 20) and the IgA was assessed in 100 μL of the diluted slurry. Serum IgA levels were measured using 100 μL of (1:500) diluted serum samples.

**In vitro induction of IgA from B cells**. Macs magnetic beads specific for B220 (Miltenyi Biotec) were used to isolate B lymphocytes. The cells were cultured at $1 \times 10^6$ cells/mL in RPMI 1640 supplemented with 20% FCS, L-glutamine, B-mercaptoethanol, Recombinant IL-4 (10 ng/mL), Anti-IgM (5 μg/mL), and Anti-CD40 (5 μg/mL). Twenty-four hours after initial plating, the cells were treated with IL-17 or TGFβ for an additional 3- or 7 days.

**Magnetic activated cell sorting for IgA-coated bacteria**. 10% fecal suspension was prepared in PBS and then centrifuged at $50 \times g$ for 15 min at 4 °C) to remove food particles. Fecal bacteria were pelleted by centrifugation at $8000 \times g$ for 5 min at 4 °C and then resuspended in 1 mL staining buffer. Fecal bacteria were stained with PE-conjugated IgA (Catalog # 12-4204-82, eBioscience) for 30 min and then sorted using anti-PE magnetic activated cell sorting (MACS) beads for 15 min at 4 °C. The IgA-coated bacteria were flow-sorted and Bacterial DNA was then isolated ($2 \times 10^6$) and then qPCR was performed using bacteria-specific primers listed in Supplementary table 1.

**Flow cytometry**. Peyer's patches and colon tissues were collected and thoroughly washed with cold 1× PBS. To obtain epithelial cells from Peyer's patches, tissues were incubated with dissociation medium (0.5 mM EDTA in RPMI) for 20 min at 37 °C. The dissociated cells were collected by filtering the supernatant through a 40 μm nylon mesh and then washed with RPMI medium. Single cell suspension from colon was prepared by digesting with 0.5 mg/mL collagenase in RPMI medium for 30 min at 37 °C with vigorous shaking. The supernatant was filtered and washed with RPMI medium and the single cells were then ready for antibodies staining. For T cell and B cell staining, APC eFluor780-conjugated anti-CD45 (Catalog #47-0451-82), PE-Cy7-conjugated anti-CD4 (Catalog #25-0041-82), APC-conjugated anti-Foxp3 (Catalog #17-5773-82), FITC-conjugated anti-CD3e (Catalog #11-0031-81), PE-conjugated anti-CXCR5 (Catalog #12-7185-80), and eFluor450-conjugated anti-PD-1 (Catalog #48-9981-80) from eBioscience (San Diego, CA) or Alexa Fluor647-conjugated anti-B220 (Catalog #103229) from BioLegend (San Diego, CA) were utilized. For PE-conjugated anti-IL17A (Catalog #506903, BioLegend, San Diego, CA) staining, cells were treated with phorbol myristate acetate (PMA, 50 ng/mL; LC laboratories, Woburn, MA), ionomycin (500 ng/mL; LC laboratories) and 1× Brefeldin A (eBioscience) for 4 h. In addition, to perform the intracellular staining for cytokines FoxP3 and IL17A, cells were fixed and permeabilized with 1× permeabilization buffer (eBioscience) following cell surface staining according to the manufacturer's protocol. Flow cytometry analysis was performed using the BD LSRFORTESSA X-20 (BD Biosciences) instrument and the data were analyzed with FlowJo 10.2 software (FlowJo LLC, Ashland, OR). For the gating strategy, CD4 T cells were gated on single cells, CD45+CD3+CD4+ cells, the subsets were then further gated with Foxp3+ as T-reg cells, GFP+ (RORC(γτ)-EGFP) as Th17 cells, PD1+CXCR5+ as T-follicular cells. B cells were gated with single cells, then CD45+B220+ cells. Flow-assisted cell sorting was completed using the MoFlo Astrios I (Beckman Coulter, IN). Gating was done on cells for APC-eflour 780-CD45, FITC-CD3, and PEcy7-CD4 positivity (CD45+CD3+CD4+). Cells were suspended in HBSS+ 2% FBS and ~$1 \times 10^5$ cells were sorted directly into TRIzol LS reagent (ThermoFisher Scientific, MA).

**Statistical analysis**. Sample sizes were chosen based on previous experience and known variation with the colitis and septic models. For Peyer's patches analysis, the N number in the figure legends denotes the number of mice used per group. Wherever the data is presented as bar graph, the animal numbers were listed in the figure legends. Results are expressed as mean ± SEM. Significance among multiple groups was tested using one-way analysis of variance and significance between the two groups were calculated by Student's $t$-test. For survival assays, comparisons were performed by a Log-rank test.

**Reporting summary**. Further information on experimental design is available in the Nature Research Reporting Summary linked to this article.

## Data availability

The microarray data have been deposited in GEO database under the accession number GSE86194. The 16S sequencing data have been deposited in Sequence Read Archive (SRA) database under the accession numbers SAMN10753948-SAMN10753962 (e.g. see https://www.ncbi.nlm.nih.gov/biosample/SAMN10753948/). The RNA data have been deposited in SRA database under the accession number SRP109133. The source data underlying Fig. 4a and Supplementary Fig 4e and uncropped blots are provided as a Source Data File.

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

## Acknowledgements

This work was supported by NIH grants (CA148828 and DK095201), Department of Defense grant (CA171086), a pilot grant from the University of Michigan GI Spore (CA130810) and University of Michigan Medical School Host Microbiome Initiative (HMI) Microbiome Explorer Program (MEP) (Y.M.S.). S.K.R. was supported by a postdoctoral fellowship from AHA (15POST22650034) and K99 from NIDDK (K99DK110537). X.X. was supported by Research Scholar Award from the American Gastroenterological Association. D.T. and A.J.S. were supported by a T32 training grant (DK 094775 and GM 008322). J.A.C. was supported by NIH grant (R01ES028802). L.R. was supported by NIH grant (DK091591). NIH grants T32HL007517 and 5P30DK034933 (to M.Y.Z.), and DK095782 (to G.N.).

## Author contributions

S.K.R., H.Z., and Y.M.S. Conceived and designed the study. I.J., S.N.D., and X.M. performed flow analysis. A.J.S. performed enteroid experiments. D.T. and X.X. analyzed the results. N.K.D. performed and analyzed 16S sequencing. M.Y.Z., M.C., and G.N did IgA analysis. Y.H., R.M.M., C.E.W., and L.Y.R. provided animal models. J.K.G. did pathological scoring. J.A.C. did RNA-SEQ analysis. S.K.R. and Y.M.S. analyzed the results and wrote the manuscript.

## Additional information

**Competing interests:** The authors declare no competing interests.

