## [Peer Review File · Nature Communications]

Reviewers' comments:

Reviewer #1 (Innate-adaptive crosstalk, microbiota)(Remarks to the Author):

Ramakrishnan et al. examined the role of non-canonical NFκB signaling in the intestinal epithelial cells. The authors demonstrate in this paper that the non-canonical NFκB signaling is important in the maintenance of M cells and that the non-canonical NFκB signaling plays an important role in the luminal sensing of commensal microbes and systemic IL-17A and IgA production. The authors showed that both impaired non-canonical NFκB signaling and constitutive non-canonical NFκB signaling lead to the susceptibility to intestinal inflammatory responses. Impaired non-canonical NFκB signaling resulted in low level of IL-17 and IgA and the reduction of IgA coated commensal microbes. Interestingly, constitutive non-canonical NFκB signaling induced ectopic M cell differentiation and a chronic increase in IL-17A production.

The experiments were well designed and performed in a logical manner. The experimental results support the authors' conclusions. There are, however, several points that need to be clarified.

1) The authors demonstrated that the constitutive non-canonical NFκB signaling induced ectopic M cell differentiation. It is intriguing that constitutive non-canonical NFκB signaling did not alter M cells in Peyer's patches and colonic LF yet induced ectopic M cells. Is there is any preference in the location of ectopic M cell differentiation?

2) Results shown in Fig. 5b demonstrated that IL-17 enhanced IgA production. Lin et al. (PMID: 29299950) showed that IL-17 stimulates IgA production by an IgA positive B cell line. Does IL-17 induce class switch recombination or simply induces plasma cell differentiation of IgA positive B cells in polyclonal B cells as observed in a B cell line by Lin et al.? Is the mechanism(s) known?

3) On page 10, the authors stated that pathway analysis revealed that RANK-NIK signaling regulates key inflammatory pathways and cited Supplementary Fig. 4e. However, there is no difference in chemokine signaling and cytokine-cytokine receptor interaction in this figure. Differences are found in glycerolipid metabolism, mineral absorption, pancreatic secretion, and PPAR signaling pathway. Which pathway do the authors call the key inflammatory pathway?

4) On page 11, the authors stated that M-cell differentiation does not increase IL17a mRNA levels in the enteroid and cited Supplementary Fig. 3b. Is this the right figure to be cited?

5) In Supplementary Fig. 8, panels "b" and "c" should be "c" and "b", respectively.

Reviewer #2 (IgA, gut epithelial) (Remarks to the Author):

In this manuscript the authors have analyzed the role of non classical NF-κB in colitis induction. They found that NIK is involved in M cell differentiation (or maturation). This leads to an increased susceptibility to colitis and reduction in IL-17 and IgA production.

Although the authors performed a huge amount of data with different mice ko for several factors, the message of the manuscript is confused and the data often interpreted without clear support by the results. The authors should have concentrated more on analysing the phenotype of NIK epithelial specific KO trying to better understand the effect of NIK deletion. The manuscript is a collection of correlative data that may or may not be linked to lack of M cell differentiation, or to the reduction of IL-17, IL-22 or AhR.

In general, all figures related to PP enumeration and size should be performed on swiss rolls to show PP reduction in numbers and dimension. In addition, in Fig. 1e and all the others of M cell enumeration, a statistic should be reported on several PP from WT and KO mice.

Fig. 1d. Is there a reduction in villi length especially in the small intestine of *Nik Δ IE* mice?

In Fig. 1K the decrease in IgA may be due to a reduction in signaling rather than sampling by M cells. Sampling of antigens or IgA coated bacteria has not been performed.

All of the analysis carried out by the authors was on an epithelial specific KO (either NIK, IKKa or rank) which lack M cells, however, the effect observed may be due to epithelial signaling of non classical NF- κ B or related molecules rather than to the absence of M cells. To make sure that the observed effect is due to lack of M cells, they should have tried GP2-specific KO of NIK or the others molecules. Indeed, the observation that there are differentially expressed genes in conditional KO mice would suggest also a contribution of epithelial cells and not only of M cells. Further, the analysis of RNAseq was performed at steady-state, when mice are apparently normal. What would be the expression after DSS challenge?

I am puzzled by the increase of proinflammatory genes in PP of NIK KO cells. The authors in Suppl. Fig. 2 show no change in major inflammatory cytokines, but clearly there is an increase of the inflammatory score. How can these two observations be reconciled?

Fig. 4e the amount of IL-17+ T cells and all the other immune cells should be shown also as absolute numbers, to take into account the difference in size of PP.

It is not clear why authors have concentrated on IL-17 only as also AHR KO mice have different susceptibility to colitis and IL-22 is involved in fostering barrier properties.

Fig. 5. Purification of B cells with B220 does not exclude a possible contamination with plasmacytoid DCs which might mediate the effect of IL-17.

The experiment of Fig. 5 c is not clear. Why should the B cells from IL17 or NIK KO not complement the phenotype as WT B cells? I thought the effect was dependent on absence of signal by epithelial cells not B cells. If B cells are also involved, this tells that NIK or IL17 may be involved in several other processes that the authors cannot control in the epithelial specific ko.

The data on microbiota also is quite puzzling as mice with a defect either in IgA or the poly Ig receptor have a different microbiota.

The finding that antibiotic treatment partially ameliorates colitis is not surprising as DSS colitis is dependent on the microbiota.

The reduction of IgA-coated pathogenic bacteria should reduce colitis rather than increase it, particularly because the same bacteria were not found increased in a free- non IgA-coated form, if I understand correctly the meaning of Fig. 6D. Further the authors claim that cohousing of mice does not transfer the disorder, how can colitogenic bacteria be increased?

The authors have not performed a crucial experiment of testing intestinal permeability, which might be increased as a consequence of reduced IL-22 and IgAs.

As an example of misinterpretation of the data, the authors claim: 'Antigen sampling via M-cells is critical for regulating the expression of IL17 on T-cells and gut humoral response.' However, they have never shown lack of antigen sampling.

Further: 'Here, we provide direct evidence using mouse models of colitis that M-cells are essential in the regulation of the pro-inflammatory IL17A in the intestinal LF.' This also is not correct as the observed effect may only partly be dependent on lack of M cells, but may be due to a reduction in anti-inflammatory genes, IL-22 and AhR signalling in response to epithelial cell NIK rather than absence of M cells.

In conclusion, while the results may be interesting, they are not conclusive and the data is very confused and often misinterpreted.

Minor points:

I could not find how the authors isolated colon LF for RNA analysis.

In general it is not clear how many times the experiments were performed and how many replicates have been used.

Reviewer # 1:

The experiments were well designed and performed in a logical manner. The experimental results support the authors' conclusions. There are, however, several points that need to be clarified.

We thank the Reviewer for the comments and we have addressed all the concerns raised by the Reviewer.

1) The authors demonstrated that the constitutive non-canonical NF κ B signaling induced ectopic M cell differentiation. It is intriguing that constitutive non-canonical NF κ B signaling did not alter M cells in Peyer's patches and colonic LF yet induced ectopic M cells. Is there any preference in the location of ectopic M cell differentiation?

We observed ectopic M-cells only in the colon but not in the small intestine epithelium (Data added qPCR for M marker in SI of NikVilRosaT3, Supplementary Fig. 2h). Investigation is underway to determine the mechanism involved in preferential differentiation of M-cells in colon.

2) Results shown in Fig. 5b demonstrated that IL-17 enhanced IgA production. Lin et al. (PMID: 29299950) showed that IL-17 stimulates IgA production by an IgA positive B cell line. Does IL-17 induce class switch recombination or simply induces plasma cell differentiation of IgA positive B cells in polyclonal B cells as observed in a B cell line by Lin et al.? Is the mechanism(s) known?

Our goal was to show there is a direct link between IL17 and B-cells, and we thank you for pointing out this interesting manuscript, which further provides evidence. We also show loss of NIK signaling decreased the expression of genes involved in class switching such as Ighv1-47, Igkv1-133 and Igkv9-120 in the NikKO mice (Supplemental figure S4d). However, in the Peyer's patches of IL17KO mice Igkv1-133 but not Ighv1-47 and Igkv9-120 were changed (Supplementary Fig. 5d). This suggests that IL17 may not directly affect the class switching but rather regulate antigen-dependent B-cell differentiation.

3) On page 10, the authors stated that pathway analysis revealed that RANK-NIK signaling regulates key inflammatory pathways and cited Supplementary Fig. 4e. However, there is no difference in chemokine signaling and cytokine-cytokine receptor interaction in this figure. Differences are found in glycerolipid metabolism, mineral

absorption, pancreatic secretion, and PPAR signaling pathway. Which pathway do the authors call the key inflammatory pathway?

We apologize for the confusion. The pathway analysis are graphed so controls were normalized to 1 and the bars represent the difference in the Rank or NIK KO compared to the controls. We have added the detail in the figure legend (Supplementary Fig. 4e).

4) On page 11, the authors stated that M-cell differentiation is not increase Il17a mRNA levels in the enteroid and cited Supplementary Fig. 3b. Is this the right figure to be cited? The figure S3b is correctly cited in the original submission. In order to make it easier, we have now graphed the IL17 mRNA data separately and added to the corresponding figure (Supplementary Fig. 4i).

5) In Supplementary Fig. 8, panels “b” and “c” should be “c” and “b”, respectively. We have corrected the mistake in the supplement figure 8.

Reviewer #2:

In general, all figures related to PP enumeration and size should be performed on swiss rolls to show PP reduction in numbers and dimension. In addition, in Fig. 1e and all the others of M cell enumeration, a statistic should be reported on several PP from WT and KO mice.

We attempted to perform the PP size enumeration using the Swiss roll but it was highly variable due to sectioning. Based on the depth of the section through the Peyer’s patches, the sizes were variable. Moreover, based on the orientation and the tightness of the swiss sections the numbers could also be highly variable, which made it technically challenging to get any conclusive data. We believe that measuring the gross size of the Peyer’s patches using a Vernier calipers is more accurate than the swiss roll. Based on our effort to address the Reviewers comments. The bar graphs represented in the M-cell analysis are data assessed from several Peyer’s patches from the WT and KO. The N-number indicates the number of mice used rather than the number of Peyer’s patches. We have now included the description in the figure legends.

Fig. 1d. Is there a reduction in villi length especially in the small intestine of Nik Δ IE mice?

We did not observe any difference in the villi length in Nik Δ IE mice (Supplementary Fig. 2b).

In Fig. 1K the decrease in IgA may be due to a reduction in signaling rather than sampling by M cells. Sampling of antigens or IgA coated bacteria has not been performed.

We now have now included data for antigen sampling using microbeads which shows that the sampling is blunted in the Peyer’s patches of Nik Δ IE mice (Fig. 1g and Supplementary Fig. 2i).

All of the analysis carried out by the authors was on an epithelial specific KO (either

NIK, IKK α or rank) which lack M cells, however, the effect observed may be due to epithelial signaling of non-classical NF- κ B or related molecules rather than to the absence of M cells. To make sure that the observed effect is due to lack of M cells, they should have tried GP2-specific KO of NIK or the others molecules. Indeed, the observation that there are differentially expressed genes in conditional KO mice would suggest also a contribution of epithelial cells and not only of M cells. Further, the analysis of RNAseq was performed at steady-state, when mice are apparently normal. What would be the expression after DSS challenge?

The Reviewer raises a valid point to use GP2-specific Cre to dissect the contribution of epithelial vs M-cell specific loss of NIK signaling in the phenotype we observed. Unfortunately, to our knowledge, M-cell specific Cre mice are not available to perform the experiments suggested. Moreover, other models of M-cell defects such as the GP2-KO mice have their own caveats as GP2 is highly expressed on immune cells and could alter their functions. Please read above to our note to the editor of what we have done to increase the empirical evidence of our concluded pathway. In addition to the new data, we have added to the discussion the possibility of non-M-cell mediated mechanisms and the need for Cre lines to target to M-cells.

We performed RNAseq under steady state to determine the basal changes by the loss of epithelial NIK without any confounding effect of exacerbated colitis. We have added a detailed time course of histological changes following intestinal epithelial NIK disruption (Supplementary Fig. 2g). This data coupled with our inflammatory mediator expression analysis demonstrates that changes in pro-inflammatory gene expression are increased by NIK disruption when histological changes are observed, therefore pro-inflammatory changes are not an early causative mechanism of increased susceptibility of epithelial NIK disruption. Moreover, we now show that before any histological changes following DSS, epithelial barrier changes are not different between wild-type and NIK knockout mice (Supplementary Fig. 2e, f). This demonstrates that barrier dysfunction is not the early causative mechanism of increased susceptibility of epithelial NIK disruption.

I am puzzled by the increase of proinflammatory genes in PP of NIK KO cells. The authors in Suppl. Fig. 2 show no change in major inflammatory cytokines, but clearly there is an increase of the inflammatory score. How can these two observations be reconciled?

We do not see an inflammatory gene expression change in whole colon until we see a histological change and therefore do not believe NIK is directly altering cytokines or pro-inflammatory mediators (Supplementary Fig. 2-i). We now included the inflammation score of the colon during the progression of colitis assessed by the pathologist.

Fig. 4e the amount of IL-17+ T cells and all the other immune cells should be shown also as absolute numbers, to take into account the difference in size of PP.

We have added the percentage of IL17+ cells in the Fig.4 g, h.

It is not clear why authors have concentrated on IL-17 only as also AHR KO mice have different susceptibility to colitis and IL-22 is involved in fostering barrier properties.

Although, the AHR and IL22 are decreased in the NIK KO mice, we did not see any difference in the AHR expression by conventionalizing the germ-free mice with gut microbiota from WT mice at the time points we assessed. We agree with the Reviewer that apart from numerous roles of IL22 in inflammation, IL22 plays an important role in barrier function. However, we did not see any difference in the barrier function (Supplementary Fig. 2e, f). Based on these data, we rationalized to investigate if IL17 is involved in the colitis susceptibility and consistent with previous reports and combining various genetic models we propose that IL17 is partly involved in the colitis susceptibility in NIKKO mice. We have included these details in the discussion.

Fig. 5. Purification of B cells with B220 does not exclude a possible contamination with plasmacytoid DCs which might mediate the effect of IL-17.

Our goal was to show there is a direct link between IL17 and B-cells, but we do agree with the Reviewer that there may be plasmacytoid DC contamination with our preparation. However, as Reviewer 1 pointed out, literatures have suggested that IL-17 can directly induce B-cell differentiation. We have included the citations in the revised manuscript.

The experiment of Fig. 5 c is not clear. Why should the B cells from IL17 or NIK kO not complement the phenotype as WT B cells? I thought the effect was dependent on absence of signal by epithelial cells not B cells. If B cells are also involved, this tells that NIK or IL17 may be involved in several other processes that the authors cannot control in the epithelial specific ko.

We believe one of our main points is to show epithelial NIK in the M-cells alters immune cell response leading to local and systemic effects. As mentioned above this could be due to other epithelial cell type but there is no evidence in the literature to date. Specifically, to the question this is a very acute treatment of B-cells and therefore they maintain their phenotype from the donor mice. We are glad the Reviewer brought up the clarity and we have changed the result section in hopes to be clearer.

The data on microbiota also is quite puzzling as mice with a defect either in IgA or the poly Ig receptor have a different microbiota.

The Reviewer is referring to complete KO of IgA, our mice just have lower levels which may lead to less of an effect on microbiota. Moreover, our data is consistent with previous work showing no global change in microbiota in M-less mice (PMID: 26601902).

The finding that antibiotic treatment partially ameliorates colitis is not surprising as DSS colitis is dependent on the microbiota.

We agree with the Reviewer, however we show complete rescue in the ViERT2NIK mice after antibiotic treatment demonstrating a role for the microbiota.

The reduction of IgA-coated pathogenic bacteria should reduce colitis rather than increase it, particularly because the same bacteria were not found increased in a free- non IgA-coated form, if I understand correctly the meaning of Fig. 6D. Further the authors

claim that cohousing of mice does not transfer the disorder, how can colitogenic bacteria be increased?

It has been shown that IgA coating of the bacteria decreases its pathogenicity (PMID: 27562257). Therefore, decrease in the IgA coating should potentially increase the predisposition to colitis. Co-housing helps to normalize the microbiota between the mice. If the NIK KO mice are prone to colitis due to microbiota difference, then we expect that co-housing will mitigate the difference in the colitis susceptibility in NIK KO mice. However, co-housed NIKKO mice were still more susceptible to colitis suggesting that host factor is involved (in this case decreased intestinal IgA) plays a critical role in colitis predisposition.

The authors have not performed a crucial experiment of testing intestinal permeability, which might be increased as a consequence of reduced IL-22 and IgAs.

We have performed the intestinal permeability test using FITC-dextran. Interestingly, we did not see any difference in the barrier function between the WT and NIK KO mice early after DSS treatment. Moreover, the expression of occludin and E-cadherin was not different in NIK KO mice at steady state (Supplementary Fig. 2e, f).

As an example of misinterpretation of the data, the authors claim: ‘Antigen sampling via M-cells critical for regulating the expression of IL17 on T-cells and gut humoral response.’ However, they have never shown lack of antigen sampling.

We have included the data for lack of antigen sampling using microbeads (Fig. 1g).

Further: ‘Here, we provide direct evidence using mouse models of colitis that M-cells are essential in the regulation of the pro-inflammatory IL17A in the intestinal LF.’ This also is not correct as the observed effect may only partly be dependent on lack of M cells, but may be due to a reduction in anti-inflammatory genes, IL-22 and AhR signalling in response to epithelial cell NIK rather than absence of M cells. In conclusion, while the results may be interesting, they are not conclusive and the data is very confused and often misinterpreted.

We did not observe any difference in inflammatory genes unless mice were induced with DSS. Moreover, based on the data from conventionalization of germ-free mice, bone marrow transplantation experiment and barrier function assessment we believe that IL-17 is involved in the colitis phenotype of NIKKO mice. IL17 cross talk with IL22 signaling is essential for humoral immunity, so we cannot completely exclude that IL22 or other key cytokines including AHR is not involved in our inflammatory phenotype including IL17 expression. We have now included these details in the discussion section in the revised manuscript.

Minor points:

I could not find how the authors isolated colon LF for RNA analysis.

We have included a photograph and methodology to locate and isolate colon lymphoid follicles (Supplementary Fig. 1d).

In general it is not clear how many times the experiments were performed and how many replicates have been used.

We have included the sample size in the figure legends. In general, at least 1-3 Peyer's patches were analyzed per mice for RNA analysis. The n number in the figure legends are the number of mice rather than the number of Peyer's patches assessed. Wherever the n numbers are less than 5, the graphs are represented as dot graphs rather than bar graphs.

Reviewers' comments:

Reviewer #1 (Remarks to the Author):

Ramakrishnan et al. have revised their paper with additional data, which significantly improved the quality of this paper.

There are, however, a few minor points that need to be addressed.

1) Disruption of epithelial NIK for two weeks decreased the PP size and number (page 7) but disruption of epithelial RANK resulted in no difference in PP size and number (page 10). If NIK is downstream of RANK, one expects similar results in these two mouse lines. It is better to provide readers with some explanations.

2) The authors showed differential expression of genes involved in B-cell antibody class switching in *Nik^{<DletalE>}* mice. The authors should comment on the expression of *Acida* gene encoding AID, a critical gene for class switching even if there is no change in its expression.

3) Line 321 "Intestinal RANK and NIK signaling decreased serum IgA (Fig. 7e, f)" seems opposite to the observation. It seems "Loss of intestinal RANK and NIK signaling decreased serum IgA (Fig. 7e, f).

Reviewer #2 (Remarks to the Author):

I am still not very convinced about the interpretation of most of the data.

First of all the authors still did not respond to several of my concerns.

The data shown on microbeads is just one panel and there is no quantification.

Again the authors infer conclusions without having a prove: Saying: 'critical role of epithelial NIK signaling in antigen sampling via induction of M-cells' is not correct because the authors do not know why *NIK^{-/-}* mice do not have M cells, they might just die because of absence of NIK signaling.

Another conclusion which is not supported by the data:

'Here, we show that the loss of intestinal epithelial NIK signaling (*Nik^{ΔIE 126}*) is sufficient to decrease gut humoral response due to loss of antigen sampling M-cells' How can the authors claim this? They have lack of signaling also in epithelial cells, it may be due to lack of sampling but this has not been demonstrated. The authors should analyse an antigen-specific response to a given antigen to make this claim.

The authors also claim that the inflammatory response is not responsible for the early events of colitis based on three proinflammatory cytokines at RNA levels. They should have tested both the protein level and the expression of IL-10 which is the most important anti-inflammatory cytokine.

Are M cells protective, or are IgA protective? As the authors show that these mice have reduced IgA how can they claim that the effects are due to lack of M cells? Again they cannot exclude a contribution by epithelial cell expression of NIK. Regarding RankL this is fundamental for many other immune cells, that the two mice cannot be compared to support one another.

The authors quote a manuscript showing that IgA restricts colitogenic bacteria, however, another

manuscript by the group of Flavell shows that colitogenic bacteria are IgA coated (25171403).

In conclusion I still think that the authors have an interesting phenotype but that in several places their interpretation is not correct or is pushed without clear support by the data.

Reviewers' comments:

We thank the Editor and the Reviewers for giving us the 2nd chance to revise our manuscript. Please find the response to the comments in blue.

Reviewer #1 (Remarks to the Author):

1) Disruption of epithelial NIK for two weeks decreased the PP size and number (page 7) but disruption of epithelial RANK resulted in no difference in PP size and number (page 10). If NIK is downstream of RANK, one expects similar results in these two mouse lines. It is better to provide readers with some explanations.

No difference in the PP size or number suggest a possibility for residual NIK signaling in the PP at 2-weeks following temporal knockdown of RANK. We have included this statement in the results section, marked in red.

2) The authors showed differential expression of genes involved in B-cell antibody class switching in *Nik^{<DletalE>}* mice. The authors should comments on the expression of *Aicda* gene encoding AID, a critical gene for class switching even if there is no change in its expression.

We did not observe any difference in the *AICDA* and we have noted that in our results section (marked in red)

3) Line 321 “Intestinal RANK and NIK signaling decreased serum IgA (Fig. 7e, f)” seems opposite to the observation. It seems “Loss of intestinal RANK and NIK signaling decreased serum IgA (Fig. 7e, f).

We have corrected the mistake in the revised manuscript.

Reviewer #2 (Remarks to the Author):

We thank Reviewer 2 for their assessment of our work. We believe there are fundamental issues with our interpretations as we are interpreting the data as a M-cell specific defect. However, since we do not have a M-cell specific Cre, our results could be a contribution of M-cell loss or loss of NIK signaling in other epithelial compartments. We agree and have changed the manuscript throughout to reflect this. Below are the specific points that we addressed from Reviewer 2.

I am still not very convinced about the interpretation of most of the data.

First of all the authors still did not respond to several of my concerns.

The data shown on microbeads is just one panel and there is no quantification.

We have included the quantification for the microbeads (Figure ?). We agree with the Reviewer that in the absence of data from the M-cell specific loss of NIK signaling it is difficult to claim the direct link between M-cell link and IgA response, as epithelial NIK may partly contribute to the phenotype we observed. We have included this in our discussion and we have changed our interpretation in the results and the discussion section of the manuscript. The changes have been marked in red.

Again the authors infer conclusions without having a prove: Saying: ‘critical role of epithelial NIK signaling in antigen sampling via induction of M-cells’ is not correct

because the authors do not know why NIK^{-/-} mice do not have M cells, they might just die because of absence of NIK signaling.

We agree with the Reviewer. We show that M-cell differentiation is significantly attenuated by the loss of epithelial NIK using enteroids. Currently it is not known if M-cell turnover is similar to intestinal epithelial turn over rate. We show that M-cells are lost within 2 weeks after disruption of epithelial NIK/RANK partly suggesting a possibility for M-cell apoptosis in the absence of epithelial NIK/RANK. We have included this in the discussion section and marked it in red.

Another conclusion which is not supported by the data:

'Here, we show that the loss of intestinal epithelial NIK signaling (Nik Δ IE 126) is sufficient to decrease gut humoral response due to loss of antigen sampling M-cells' How can the authors claim this? They have lack of signaling also in epithelial cells, it may be due to lack of sampling but this has not been demonstrated. The authors should analyse an antigen-specific response to a given antigen to make this claim.

We agree and we have toned down the interpretation. We show that loss of epithelial NIK decreases not only fecal IgA but also IgA coating of specific bacteria. We have changed our conclusion to "loss of intestinal epithelial NIK signaling (Nik Δ IE 126) is sufficient to decrease antigen sampling M-cells and gut humoral response". We hope the Reviewer will agree with our conclusion. We feel that antigen specific response may not add to the conclusion as this could be still due to changes in epithelial NIK signaling.

The authors also claim that the inflammatory response is not responsible for the early events of colitis based on three proinflammatory cytokines at RNA levels. They should have tested both the protein level and the expression of IL-10 which is the most important anti-inflammatory cytokine.

We have included the qPCR data for the IL-10, which do not show any significant difference at the mRNA level assessed at 3-day and 7-days after DSS treatment (Figure ?).

Are M cells protective, or are IgA protective? As the authors show that these mice have reduced IgA how can they claim that the effects are due to lack of M cells? Again they cannot exclude a contribution by epithelial cell expression of NIK. Regarding RankL this is fundamental for many other immune cells, that the two mice cannot be compared to support one another.

We agree RANKL has pleiotropic effect especially on immune cells. However, we want to emphasize that we use intestinal epithelial specific RANK receptor knockout in our study and not RANKL knockout. We believe that immune cells may not be directly impacted by epithelial loss of RANK. Although we tried to prove the causal role of M-cells in colitis through IgA using bone marrow experiment in Jh^{-/-} mice, it may not completely address the M-cells vs IgA protective response. In the absence of a colitis rescue experiment with IgA in intestine specific NIK/RANK mice, we have changed our conclusion that epithelial NIK signaling is protective against colitis and speculated that this could be via M-cells.

The authors quote a manuscript showing that IgA restricts colitogenic bacteria,

however, another manuscript by the group of Flavell shows that colitogenic bacteria are IgA coated (25171403).

We agree with the Reviewer. IgA coating of the bacteria correlates with the magnitude of inflammation and IgA coating of the pathogens protects against infection through neutralization and exclusion. Here, we speculate that in the NIK and RANK knockout mice, a decrease in IgA coating of the bacteria especially the colitogenic bacteria may increase the virulence of these pathogens to full capacity resulting in severe colitis.

In conclusion I still think that the authors have an interesting phenotype but that in several places their interpretation is not correct or is pushed without clear support by the data.

We have changed some of our interpretation in line with the observation and hope that the Reviewer will agree with our conclusion.

REVIEWERS' COMMENTS:

Reviewer #1 (Remarks to the Author):

Ramakrishnan et al. have revised their paper with additional data, which further improved the quality of this paper. In Addition, the authors modified some sentences to tone-down their claims.

There are, however, still a few minor points that need to be addressed.

1) As pointed by reviewer #2, it is difficult to say that the phenotypes are induced solely by the lack of M-cells because there is no M-cell specific Cre. The authors should be careful when discussing the effect of RANK/NIK signals. Although it is likely that antigen sampling is dependent on M-cells, survival after CLP involves not only M-cells but also various factors affected by the lack of RANK/NIK signaling in epithelial cells. The authors should use "loss of RANK/NIK signaling in epithelial cells" instead of "loss of M-cells" on pages 16 and 19.

2) Figure 8i is somehow misleading. When NIK is overexpressed, M-cell numbers were increased but there was no evidence that epithelial cells or M-cells produce IL-17 as drawn in the right panel. This should be corrected.

3) The authors stated that antigenic stimulation of TLR on T-cells is essential for IL17 response from the results of BMT on page 12. It is likely, however, that myeloid cells such as DCs but not T cells are activated through TLR. Do authors have any evidence that TLR on T cells are critical for IL17 induction?

Shigeo Koyasu

We thank the Editor and the Reviewers for their time. Please find below the response to Reviewers comments marked in Blue.

REVIEWERS' COMMENTS:

Reviewer #1 (Remarks to the Author):

1) As pointed by reviewer #2, it is difficult to say that the phenotypes are induced solely by the lack of M-cells because there is no M-cell specific Cre. The authors should be careful when discussing the effect of RANK/NIK signals. Although it is likely that antigen sampling is dependent on M-cells, survival after CLP involves not only M-cells but also various factors affected by the lack of RANK/NIK signaling in epithelial cells. The authors should use “loss of RANK/NIK signaling in epithelial cells” instead of “loss of M-cells” on pages 16 and 19.

We agree with the Reviewer and we have changed the statement.

2) Figure 8i is somehow misleading. When NIK is overexpressed, M-cell numbers were increased but there was no evidence that epithelial cells or M-cells produce IL-17 as drawn in the right panel. This should be corrected.

We have corrected the model to fit with our interpretation.

3) The authors stated that antigenic stimulation of TLR on T-cells is essential for IL17 response from the results of BMT on page 12. It is likely, however, that myeloid cells such as DCs but not T cells are activated through TLR. Do authors have any evidence that TLR on T cells are critical for IL17 induction?

We do not have evidence that TLR on T cells are critical for IL17, therefore we corrected our statement as “TLRs are involved in the IL17 response” and have included a citation of a work that demonstrates it.